# Wheat powdery mildew resistance gene *Pm13* encodes a mixed lineage kinase domain-like protein

Huanhuan Li [1], Wenqiang Men [1], Chao Ma [1], Qianwen Liu [1], Zhenjie Dong[2], Xiubin Tian[3], Chaoli Wang [1], Cheng Liu[4], Harsimardeep S. Gill [5], Pengtao Ma[6], Zhibin Zhang[7], Bao Liu [7], Yue Zhao [1] ✉, Sunish K. Sehgal [5] ✉ & Wenxuan Liu [1] ✉

Wheat powdery mildew is one of the most destructive diseases threatening global wheat production. The wild relatives of wheat constitute rich sources of diversity for powdery mildew resistance. Here, we report the map-based cloning of the powdery mildew resistance gene *Pm13* from the wild wheat species *Aegilops longissima*. *Pm13* encodes a mixed lineage kinase domain-like (MLKL) protein that contains an N-terminal-domain of MLKL (MLKL_NTD) domain in its N-terminus and a C-terminal serine/threonine kinase (STK) domain. The resistance function of *Pm13* is validated by mutagenesis, gene silencing, transgenic assay, and allelic association analyses. The development of introgression lines with significantly reduced chromosome segments of *Ae. longissima* encompassing *Pm13* enables widespread deployment of this gene into wheat cultivars. The cloning of *Pm13* may provide valuable insights into the molecular mechanisms underlying *Pm13*-mediated powdery mildew resistance and highlight the important roles of kinase fusion proteins (KFPs) in wheat immunity.

Bread wheat (*Triticum aestivum* L., $2n = 6x = 42$) is one of the most important crops in the world and provides 20% of the total daily calories and protein for humans[1]. Wheat powdery mildew, caused by the fungus *Blumeria graminis* f. sp. *tritici (Bgt)*, is one of the most devastating diseases affecting grain yield and seed quality and can lead to an estimated yield loss-of approximately 1.07% globally[2]. The development of disease-resistant cultivars is considered the most economical and effective method for controlling this disease. A majority of the resistance (R) genes cloned to date encode nucleotide-binding domain leucine-rich repeat (NLR) immune receptors that recognize pathogen strain-specific effectors and activate effector-triggered immunity (ETI)[3]. However, race-specific resistance genes have become ineffective with the rapid evolution and emergence of new virulent isolates[4]. Recent research has shown that Triticeae contain an extended array of resistance genes that include diverse kinase domains[4–6], which are new players in plant immunity. Therefore, cloning novel types of powdery mildew resistance (*Pm*) genes could facilitate their fast-tracking in breeding programs and incorporation into polygene stacks to maximize the durability of powdery mildew resistance.

Although more than 60 official *Pm* resistance genes (*Pm1-Pm69*) have been identified in wheat and its wild relatives[7,8], so far only 16 of these genes have been cloned and characterized. Most of these

[1]The State Key Laboratory of Wheat and Maize Crop Science, College of Life Sciences, Henan Agricultural University, Zhengzhou 450002, PR China. [2]College of Agronomy, Nanjing Agricultural University, Nanjing 210000, PR China. [3]Institute of Genetics and Developmental Biology, Chinese Academy of Sciences, Beijing 100101, PR China. [4]Crop Research Institute, Shandong Academy of Agricultural Sciences, Jinan 250000, PR China. [5]Department of Agronomy, Horticulture and Plant Science, South Dakota State University, Brookings, SD 57007, USA. [6]College of Life Sciences, Yantai University, Yantai 264005, PR China. [7]Key Laboratory of Molecular Epigenetics of the Ministry of Education (MOE), Northeast Normal University, Changchun 130024, PR China. ✉e-mail: zhaoyue@henau.edu.cn; sunish.sehgal@sdstate.edu; wxliu2003@hotmail.com

isolated *Pm* genes, including *Pm3*[9], *Pm8*[10], *Pm2*[11], *Pm21*[12,13], *Pm60*[14], *Pm17*[15], *Pm41*[16], *Pm5e*[17], *Pm1a*[18], *MlIW172/MlWE18*[19], *Pm12*[20] and *Pm69*[8], encode NLR immune receptors. Only a few partial resistance genes have been identified, including *Pm38/Yr18/Lr34/Sr57* and *Pm46/Yr46/Lr67/Sr55*, which encode an ATP-binding cassette (ABC) transporter and a hexose transporter[21,22], respectively. Among the recently cloned genes, *Pm24* (*WTK3*) and *WTK4* encode tandem kinase proteins[4,6], and *Pm4* encodes a chimeric protein of a serine/threonine kinase and multiple C2 domains and transmembrane regions[5].

The genetic bottlenecks associated with wheat polyploidization and domestication, as well as selection in agroecosystems, led to a decrease in wheat genetic diversity and an increase in its vulnerability to biotic and abiotic stresses[23]. One way to improve resistance against these stresses is recruiting the adaptive potential of wild relatives of common wheat[24]. *Aegilops longissima* Schw. et Musch. (2*n* = 2*x* = 14, $S^lS^l$) is a wild diploid species in the secondary gene pool of wheat[25]. It possesses considerable genetic diversity for improving grain protein[26], tolerance to drought[27] and resistance to diseases[28–33]. *Ae. longissima* is also an important resource for powdery mildew resistance and several genes (*Pm13*, *Pm66*, and *Pm6Sl*) have been reported[28,31,32]. In addition to the dominant gene *Pm13* identified on chromosome 3S^l#1S from the *Ae. longissima* accession TL01[28,29], our previous study showed that chromosomes 3S^l#2S from accession TL20 and 3S^l#3S from accession TA1910 (TAM4) also confer powdery mildew resistance[30], the resistance genes on these chromosomes were temporarily designated *Pm13a* and *Pm13b*, respectively. Furthermore, we recently introgressed *Pm66* located on chromosome 4S^l, and *Pm6Sl* located on 6S^l from *Ae. longissima* into wheat via wheat-*Ae. longissima* recombination[31,32]. Although some of these genes (*Pm13*) were transferred to wheat more than 30 years ago, none of those genes have been cloned to date. This limits the understanding of their molecular basis for conducting disease resistance and deployment in wheat breeding via functional molecular marker-assisted selection (MAS) or as multigene cassettes.

In this work, we clone the powdery mildew resistance gene *Pm13* using a strategy that combines classical map-based cloning with *ph1b*-induced homoeologous recombination and RNA-seq of the *Pm13a* donor parent TA3575 inoculated with the *Bgt* isolate E09. *Pm13* encodes a mixed lineage kinase domain-like (MLKL) protein with an N-terminal-domain of MLKL (MLKL_NTD) domain in N-terminus and a C-terminal serine/threonine kinase (STK) domain. *Pm13* orthologues are present only in the S-genome of *Aegilops*. Thus, cloning *Pm13* will

aid in elucidating the molecular mechanism of wheat powdery mildew resistance.

## Results

### Initial mapping of *Pm13a*

In our previous study, *Pm13a* was localized on the short arm of *Ae. longissima* chromosome 3S^l#2 in the Chinese Spring (CS)-*Ae. longissima* 3S^l#2(3B) disomic substitution line TA3575 from accession TL20[30]. To map the *Pm13a* gene, a total of 43 3S^l#2-specific markers were designed based on the transcriptome sequence of the *Pm13a* donor parent TA3575[34] (Supplementary Data 1). To induce recombination between *Ae. longissima* chromosome 3S^l#2 and the homoeologous group 3 chromosome of common wheat, TA3575 was crossed with the CS *ph1b* mutant TA3809, which significantly promoted homoeologous recombination. The 300 BC$_1$F$_1$ plants were genotyped using the *ph1b*-specific marker ABC302.3[35] and two 3S^l#2-specific molecular markers CL79382 and CL10208, and 25 individuals monosomic for 3S^l#2 and homozygous for *ph1b* were selected and self-pollinated to produce BC$_1$F$_2$ segregating populations for developing 3S^l#2 recombinants.

A total of 1580 BC$_1$F$_2$ plants were screened using four 3S^l#2-specific markers, which included the markers CL54397 and CL88266 at the distal ends, and the markers CL1543 and CL90315 in the proximal regions of each arm of 3S^l#2. As a result, 46 plants displaying markers disassociation were selected as 3S^l#2 recombinants. These 46 recombinants were further characterized using 43 3S^l#2-specific markers and categorized into 22 different groups (Supplementary Data 2). The F$_2$ families of each recombinant were evaluated for powdery mildew resistance by inoculation with the *Bgt* isolate E09. Marker characterization of 46 3S^l#2 recombinants and powdery mildew resistance evaluation resulted in physical mapping of *Pm13a* to the 3.67 Mb terminal interval on 3S^l#2 S flanked by markers CL10208 and CL56281 (Supplementary Fig. 1). Seven markers (CL22345, CL87265, CL61058, CL897, CL51739, CL3352 and CL9539) were found to cosegregate with *Pm13a* in the mapped region (Fig. 1a, b).

### High-resolution mapping and cloning of *Pm13a*

To fine-map *Pm13a*, we developed an additional set of 14 3S^l#2-specific markers from recently released *Ae. longissima* reference sequences[36,37] (Supplementary Data 3) and used them to identify 65 3S^l#2 recombinants from self-pollinated progenies of heterozygous resistant 3S^l#2 recombinants (Supplementary Data 4). Analysis of 65 recombinants

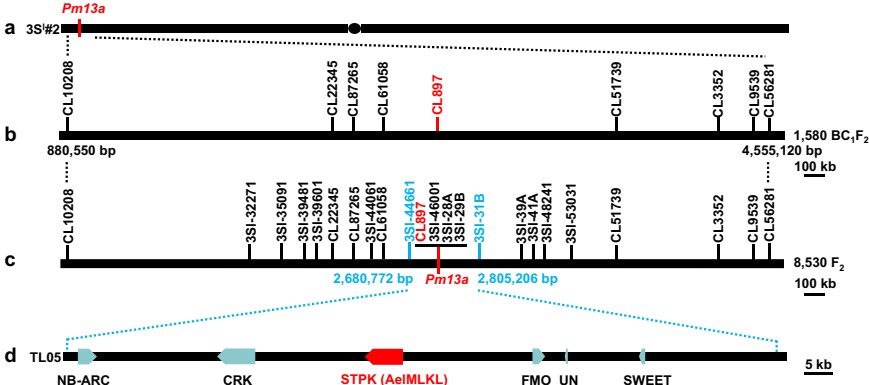

**Fig. 1 | Map-based cloning of *Pm13a*. a** *Ae. longissima* chromosome 3S^l#2. **b** Initial mapping of *Pm13a* to a 3.67 Mb genomic region flanked by markers CL10208 and CL56281 on the short arm of chromosome 3S^l#2. **c** Fine mapping of *Pm13a* to a 124.43 kb region between markers 3Sl-44661 and 3Sl-31B on the *Ae. longissima* chromosome arm 3S^l#2S. **d** *Pm13a* candidate genes predicted in the 3Sl-44661-3Sl-31B marker interval. Six genes were identified in this interval, these genes, an NB-ARC domain-containing protein (NB-NRC), a cysteine-rich receptor-like protein

kinase (CRK), a flavin-containing monooxygenase (FMO), an unknown protein (UN) and a protein that mediates both low-affinity uptake and efflux of sugar across the membrane (SWEET) were obtained from the *Ae. longissima* TL05 reference genome. A serine/threonine protein kinase (STPK) was identified from unigene CL897Contig1 in the TA3575 transcriptome sequence and was missing from the *Ae. longissima* TL05 reference genome.

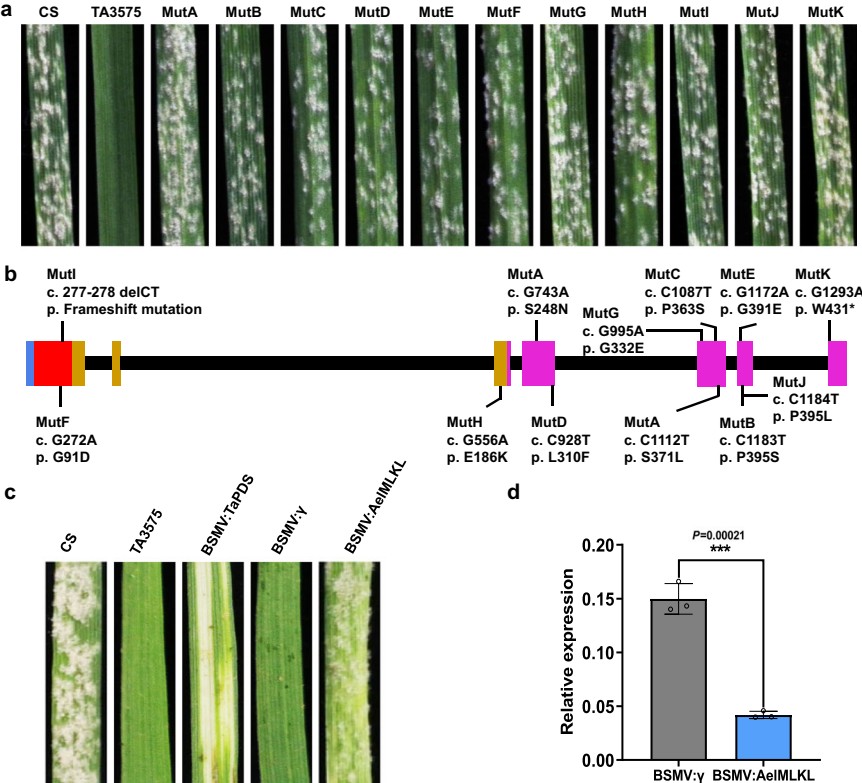

**Fig. 2 | Validation of *Pm13a* candidate gene *AelMLKL* using EMS-induced mutants and VIGS. a** Powdery mildew resistance assessments of *AelMLKL* EMS mutants. Infection phenotypes of resistant line TA3575, susceptible control CS, and 11 susceptible mutants inoculated with *Bgt* isolate E09 at 7 d post-inoculation (dpi). **b** EMS-induced mutants carrying frameshift, nonsense, and missense mutations in the *AelMLKL* gene sequence. The structure of the *AelMLKL* gene (from the start to the stop codon) is presented. Black straight lines indicate introns, and rectangles indicate coding exons (red, magenta and bright orange rectangles represent the MLKL_NTD and STK domains, and the brace region, respectively). The positions of the mutations are indicated by thin vertical lines. Mutation names in black indicate the coding sequence (c.) changes and their predicted effects on the translated protein (p.). **c** Symptoms of the third leaves of representative plants subjected to VIGS 7 days after inoculation with *Bgt* isolate E09 are shown. **d** Expression levels of the *AelMLKL* of BSMV:γ- and BSMV:*AelMLKL*-infected TA3575 plants assessed via qRT-PCR. Five leaves were used for each biological replicate, and three biological replicates were used for each group. The values of qRT-PCR are the mean ± SD (two-sided *t*-test, *n* = 3 biologically independent experiments, ***$P < 0.001$). Source data are provided as a Source Data file.

using the 14 new and seven above-mentioned markers in conjunction with the response to E09 infection resulted in the delimitation of *Pm13a* to a 124.43 kb interval flanked by markers 3Sl-44661 and 3Sl-31B in *Ae. longissima* TL05 reference genome (Supplementary Fig. 2, Fig. 1c).

The *Pm13a* interval based on TL05 reference genome[36] included only five annotated genes, TL05.3S01G0012000.1, which encodes a protein with an NB-ARC domain (*NB-ARC*); TL05.3S01G0012100.1, which encodes a cysteine-rich receptor-like protein kinase (*CRK*); TL05.3S01G0012200.1, which encodes a flavin-containing mono-oxygenase (*FMO*); TL05.3S01G0012300.1, which encodes an unknown protein (*UN*); and TL05.3S01G0012400.1, which encodes a protein that mediates both low-affinity uptake and efflux of sugar across the membrane (*SWEET*) (Fig. 1d). Furthermore, four markers (CL897, 3Sl-46001, 3Sl-28A and 3Sl-29B) cosegregated with *Pm13a* in this interval, of which only the CL897 marker, which was derived from the TA3575 transcriptome unigene CL897Contig1, was annotated as a serine/ threonine protein kinase (*STPK*) (Supplementary Data 5, Fig. 1d) and was missing from the TL05 reference genome assembly. The remaining three markers, though present in the *Ae. longissima* reference sequences were not annotated as genes. Transcriptome comparison between TA3575 and CS revealed that only *STPK* was specifically expressed in TA3575 before and after *Bgt* E09 inoculation, whereas none of the five annotated genes from the TL05 reference genome were expressed in both TA3575 and CS. Sequence alignment revealed that *STPK* was present only in TA3575 and was absent in both CS and TL05 reference genomes. Thus, the *STPK* gene was considered the best candidate for *Pm13a*.

The full-length sequence of *STPK* was cloned from TA3575 using the primers listed in Supplementary Data 6. The *STPK* gene is 6552 bp from the start codon to the stop codon and contains seven exons with a coding sequence (CDS) of 1431 bp. The gene encodes a 476-amino acid protein containing a MLKL_NTD domain in its N-terminus and a C-terminus STK domain based on the conserved domain database from the National Center for Biotechnology Information (NCBI), which possesses the same MLKL domain architecture as the vertebrate necroptosis mediator MLKL[38], thus *STPK* was re-designated as *AelMLKL*.

## Validation of *AelMLKL* by loss-of-function mutants

To determine whether *AelMLKL* was required for *Pm13a* resistance, we mutagenized 4800 seeds of TA3575 with 0.6% ethylmethanesulfonate (EMS), and 506 M₁ plants were harvested. Sixteen M₂ seedlings from each M₁ family were screened for susceptible mutants using the *Bgt* isolate E09, and twenty-five families segregating for resistance and susceptibility were identified and tested in the M₃ generation. Finally, 14 independent susceptible mutants were verified after powdery mildew resistance evaluation, 3Sl#2-specific marker analysis, and in situ hybridization identification in the M₃ generation. When comparing both cDNA and gDNA sequences of the *AelMLKL* gene from these mutants with those of TA3575, 11 out of 14 susceptible mutants had single nucleotide polymorphisms (SNPs) or deletions that resulted in amino acid substitutions, premature stop codon or frameshift muta-tion (Fig. 2a, b, Supplementary Fig. 3, Supplementary Table 1). Mutant F (MutF) had one missense mutation in the MLKL_NTD domain (p.G91D).

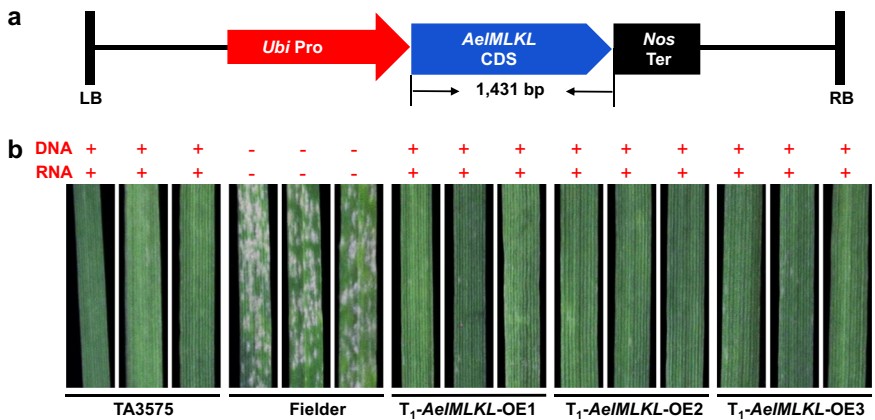

**Fig. 3 | Transgenic validation of *Pm13a* candidate gene *AelMLKL*. a** Structure of *ProUbi:AelMLKL* used for transformation of susceptible cultivar Fielder. The construct contains the gene coding sequence (CDS), the maize ubiquitin (*Ubi*) promoter, and nopaline synthase (*Nor*) terminator. LB left border, RB right border. **b** Powdery mildew responses of the resistant line TA3575 (IT 0), Fielder (IT 4), and the T₁ transgenic plants of *ProUbi:AelMLKL* (IT 0) to *Bgt* isolate E09. Three representative individuals of each transgenic line were photographed at 7 dpi; the presence (+) or absence (−) of *AelMLKL* is indicated.

MutI had a 2-bp deletion in the MLKL_NTD domain that led to a frameshift mutation. MutH had one missense mutation in the brace region between the MLKL_NTD and STK domains (p.E186K). MutK was a nonsense mutation that gave rise to premature stop codons at the encoded amino acid positions of 431 in the STK domain (p.W431*). MutA had two missense mutations within the STK domain (p.S248N and p.S371L). The remaining six independent mutants (MutB, MutC, MutD, MutE, MutG and MutJ) each had a nonsynonymous mutation (p.P395S in MutB, p.P363S in MutC, p.L310F in MutD, p.G391E in MutE, p.G332E in MutG and p.P395L in MutJ) in the STK domain. These independent mutations demonstrated that the MLKL_NTD and STK domains and the connection brace of AelMLKL are necessary for the resistance response against *Bgt*. No sequence variation was detected for the other three EMS-induced susceptible mutants in the *AelMLKL* gene, suggesting possible mutations in other unknown genes involved in the *AelMLKL* regulatory pathway.

### Silencing of *AelMLKL* compromises powdery mildew resistance in TA3575

To further validate whether *AelMLKL* mediates resistance to powdery mildew, we employed *Barley stripe mosaic-virus-induced* gene silencing (*BSMV*-VIGS) to specifically knock down the expression of the *AelMLKL* gene in TA3575. Powdery mildew symptoms and fungal sporulation pustules were observed on the leaves with reduced *AelMLKL* expression through VIGS (Fig. 2c, d), confirming that a high level of powdery mildew resistance in TA3575 required a certain level of *AelMLKL* expression.

### Transgenic expression validates *AelMLKL* as *Pm13a*

To further validate whether *AelMLKL* was sufficient to confer powdery mildew resistance, we performed transgenic expression via the CDS of *AelMLKL* driven by the maize ubiquitin (*Ubi*) promoter into susceptible wheat cultivar Fielder. As a result, 14 of the 20 transgenic T₀ plants were confirmed to have *AelMLKL* by PCR analysis and conducted immune responses to *Bgt* isolate E09 (Infection type (IT) = 0) (Fig. 3, Supplementary Table 2). Among the 14 T₁ transgenic positive plants resulting from self-pollination of different T₀ individuals, AelMLKL-OE3 displayed no significant difference in *AelMLKL* expression compared to TA3575, while the remaining 13 T₁ lines exhibited higher *AelMLKL* expression (Supplementary Fig. 4). These results indicated that both normal expression and overexpression of *AelMLKL* can effectively respond to *Bgt* isolate E09. Taken together, the results from the EMS mutagenesis, *BSMV*-VIGS, and transgenic experiments consistently support that *AelMLKL* is the functional *Pm13a* gene.

### Allelism at the *Pm13* locus

We previously reported that *Pm13* (R1B), *Pm13a* (TA3575), and *Pm13b* (TA7545) are located on the short arms of chromosome 3Sˡ from different *Ae. longissima* accessions[30]. To determine whether these three genes are the same, we first performed an allelism test by conducting reciprocal crosses with R1B (*Pm13*), TA3575 (*Pm13a*), and TA7545 (*Pm13b*), and subsequently phenotyped their F₂ individuals by inoculation with the *Bgt* isolate E09 at the seedling stage. All the F₂ plants from the crosses R1B × TA3575 (2183 plants), R1B × TA7545 (2715 plants), TA3575 × R1B (2785 plants), TA3575 × TA7545 (2176 plants), TA7545 × R1B (2082 plants) and TA7545 × TA3575 (2168 plants) displayed no segregation for powdery mildew resistance (Supplementary Table 3). Furthermore, we performed a resistance spectrum assay using 36 *Bgt* isolates with different virulence spectra collected from the main wheat-growing provinces of China, the R1B, TA3575, and TA7545 lines were immune or highly resistant to each *Bgt* isolate (IT 0-1) at the seedling stage (Supplementary Data 7). These results proved that *Pm13*, *Pm13a*, and *Pm13b* are allelic or tightly linked. Subsequently, comparing the gDNA sequences of the *AelMLKL* gene from these accessions confirmed that R1B and TA7545 had *AelMLKL* gene sequences identical to that of TA3575, thus providing strong evidence that *Pm13*, *Pm13a*, and *Pm13b* are the same gene.

### Subcellular localization and structural analysis of the Pm13 protein

To determine the subcellular location of Pm13, we fused the *Pm13* gene (CDS sequence without the terminal codon) to the N-terminus of a green fluorescent protein (GFP) reporter gene, and the resulting expression cassette (*35S::Pm13-GFP*) was transiently expressed in wheat leaf protoplasts. As shown in Supplementary Fig. 5, green fluorescence was ubiquitously detected in the cells transformed with the *35S::GFP* control. Similarly, fluorescence of the Pm13-GFP fusion protein was observed in the nucleus and cytoplasm.

To understand its detailed architecture, we used AlphaFold v2.0[39] to construct a three-dimensional (3D) model of Pm13. The 3D model revealed that the N-terminal MLKL_NTD domain of Pm13 consisted of a four-helix bundle (4HB) structure that is also defined as the HeLo domain or DUF1221 domain (Fig. 4a, b). The C-terminal STK domain of Pm13 was composed of a smaller N-lobe made up of five antiparallel β strands and the helix αC, and a larger C-lobe composed principally of α helices (Fig. 4a, c), and the two domains were connected by a three helices-containing brace (Fig. 4a). The frameshift, missense, and nonsense mutations of 11 susceptible mutants spread across the MLKL_NTD and STK domains, and their brace connection of Pm13 and

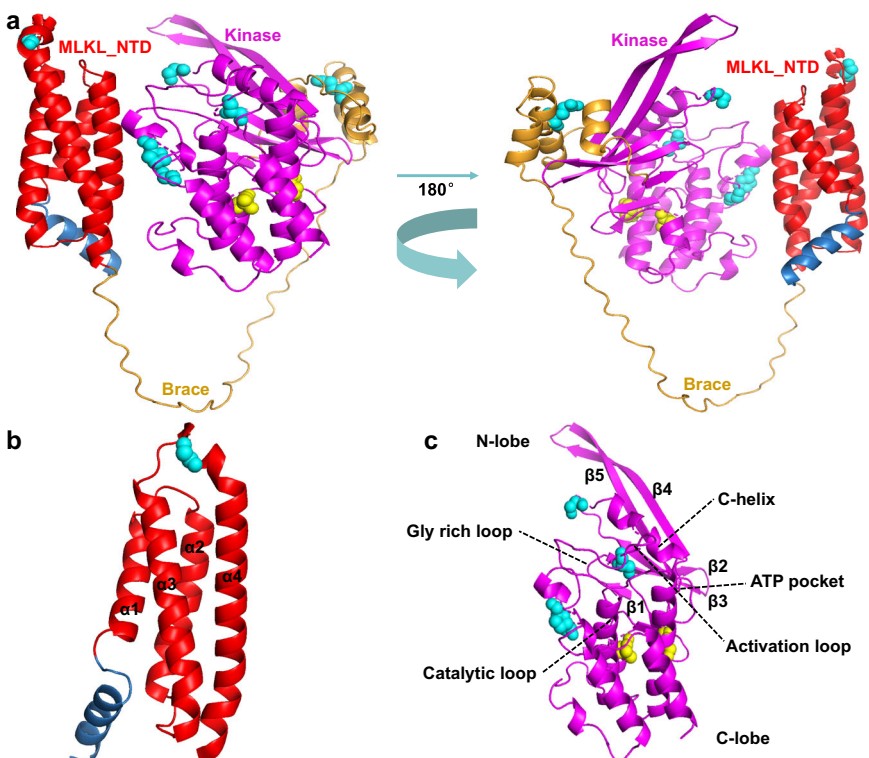

**Fig. 4 | Protein structure prediction and mutation analysis of Pm13. a** Tertiary structure of Pm13 predicted by AlphaFold v2.0. **b** MLKL_NTD domain predicted by AlphaFold v2.0; the MLKL_NTD domain consists of a four-helix bundle (4HB) structure. **c** STK domain predicted by AlphaFold v2.0, indicating the basic secondary functional features. Red: MLKL_NTD domain; Magenta: STK domain; Bright orange: Brace. Yellow spheres indicate internal amino acid substitutions resulting from the EMS mutagenesis. Cyan spheres represent surface-localized amino acid substitutions.

resulted in loss-of resistance to powdery mildew, indicating that both the domains and connection brace might be required for Pm13 to confer resistance against powdery mildew.

Structural homology modeling of the Pm13 protein revealed that the N-terminal MLKL_NTD domain of Pm13 superimposed well with the MLKL_NTD domain of animal MLKL[38], the DUF1221 domain of *Arabidopsis thaliana* AtMLKL[40] and the CC$_R$ domain of *Nicotiana benthamiana* NRG1.1[41] (Supplementary Fig. 6). All of these domain-containing proteins are involved in regulating cell death[38,40,42]. The C-terminal kinase domain of Pm13, in contrast to the pseudokinase domains of the mammalian MLKL and *A. thaliana* AtMLKLs, contains all the key conserved residues for kinase activity according to sequence alignment[43] (Supplementary Data 8). Pm13 is predicted to be a functional protein kinase based on the presence of key residues for kinase activity. Therefore, *Pm13* encodes a kinase fusion protein (KFP) with an unusual domain architecture that contains a C-terminus kinase domain fused to a MLKL_NTD domain in the N-terminus, this structure is obviously different from that of other cloned plant disease resistance genes.

### Phylogenetic analysis of Pm13

Mahdi et al. identified a limited number of MLKL proteins[40], and the N-terminal domain of these MLKLs was classified as the DUF1221 domain. To investigate the evolutionary relationships between Pm13 and other MLKLs, we retrieved 714 MLKL_NTD-Kinase proteins (Supplementary Data 9) and 97 DUF1221-Kinase proteins (Supplementary Data 10) in *Poaceae* and *Arabidopsis* from the Interpro database (https://www.ebi.ac.uk/interpro) and conducted phylogenetic analysis together with Pm13 and three AtMLKL proteins. These 815 proteins were classified into two distinct clusters, namely, the MLKL_NTD-Kinase cluster and the DUF1221-Kinase cluster. Pm13 belonged to the MLKL_NTD-Kinase cluster but was in a subfamily distinct from any other MLKL_NTD-Kinase proteins (Supplementary Fig. 7).

We used a Hidden Markov Model-based classification approach for protein kinases developed by Lehti-Shiu & Shiu[44] to analyze the evolution of the Pm13 kinase domain. The results showed that the Pm13 kinase domain belongs to the DLSV (DUF26, SD-1, LRR-VIII, and VWA) protein kinase subfamily in the RLK (receptor-like kinase)/Pelle family. Further investigation revealed that at least 44 proteins in *Poaceae* shared more than 60% sequence identity with the Pm13 kinase domain. The closest homolog of the Pm13 kinase domain was the cysteine-rich receptor-like kinase (CRK) encoded by the *T. dicoccoides CRK26* gene (Supplementary Data 11). Phylogenetic analysis of the kinase domains of Pm13 and those 44 homologs together with recently cloned KFPs, revealed that the kinase domain of Pm13 and 44 homologs belong to the LRR_8B subfamily (Supplementary Fig. 8), the most frequent kinase subfamily found in cloned KFPs[45].

We performed a BLASTP search using the MLKL_NTD domain of Pm13 across the Triticeae tribe in WheatOmics (http://wheatomics.sdau.edu.cn/), which identified 220 MLKL_NTD domain-containing proteins (Supplementary Data 12, 13) and 228 MLKL_NTD domains (Supplementary Data 14). The phylogenetic trees revealed that Pm13 and three MLKL_NTD-Kinase proteins from the S-genome of *Aegilops* species were in the same clade (Supplementary Figs. 9, 10). Among them, TB01.3S01G0016800.1 from *Ae. bicornis* TB01, TH02.3S01G0013600.2 from *Ae. sharonensis* TH02 and TS01.3B01G0036300.1 from *Ae. speltoides* TS01 shared 99.37%, 97.27%, and 89.94% protein sequence identity with Pm13, respectively (Supplementary Table 4, Supplementary Fig. 11). Further synteny analysis using five *Ae. longissima* genes on either side of *Pm13* across the Triticeae tribe demonstrated that these three genes were orthologous to *Pm13* (Supplementary Fig. 12). These findings suggest that the Pm13 architecture may be present only in the S-genome of the *Aegilops* genus and that the *Pm13* gene may have arisen after the origin of the S-genome.

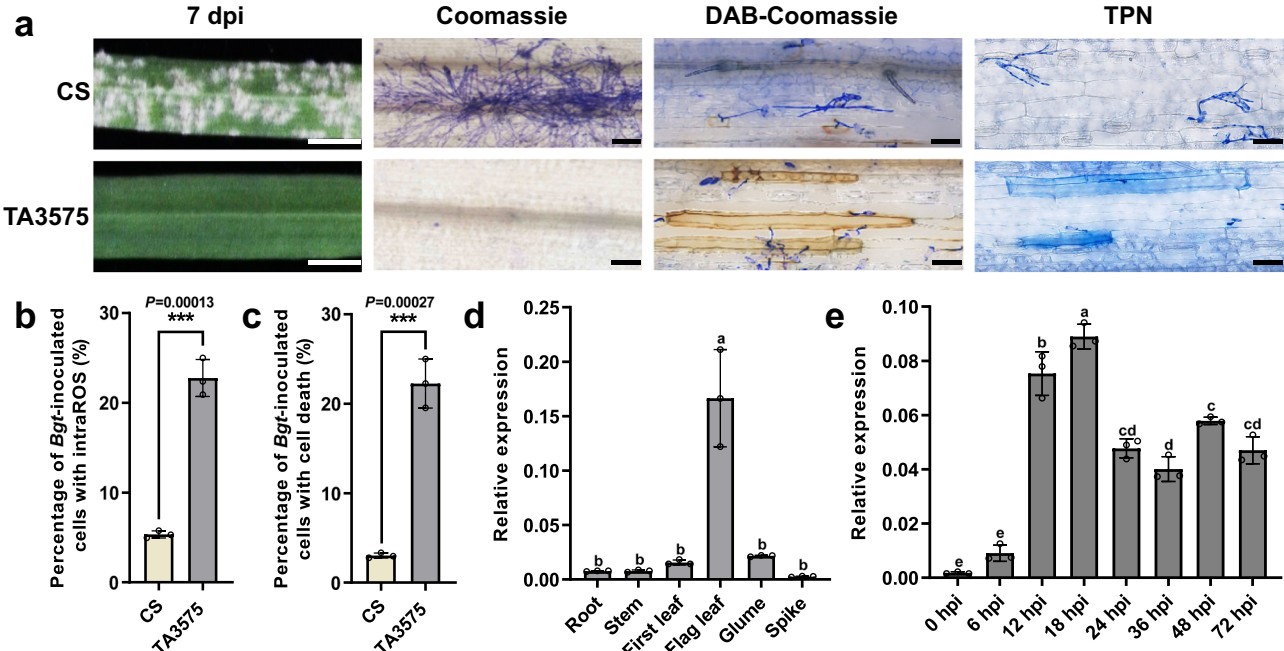

**Fig. 5 | Phenotype and expression patterns of *Pm13* after *Bgt* inoculation.**
**a** Macroscopic and microscopic characterization of TA3575 and the susceptible control CS after inoculation with *Bgt* isolate E09. From left to right: Macroscopic view of the infected leaf segments at 7 dpi. Scale bar, 5 mm. Coomassie blue staining of infected leaves at 7 dpi to visualize fungal structures. Bar, 10 μm. DAB-Coomassie blue staining of infected leaves at 48 hpi. Brown staining showed ROS accumulation. Bar, 10 μm. Trypan blue staining of the infected leaves at 48 hpi to visualize cell death. Bar, 10 μm. **b** Percentage of *Bgt*-inoculated cells with intraROS. The values are the mean ± SD (two-sided *t*-test, *n* = 3 biologically independent experiments, ***P < 0.001). **c** Percentage of cells with cell death of *Bgt*-inoculated cells. The values are the mean ± SD (two-sided *t*-test, *n* = 3 biologically independent

experiments, ***P < 0.001). **d** Transcript levels of *Pm13* in different tissues of TA3575, including the first leaves at three-leaf stage and roots, stems, flag leaves, glumes, and spikes at the heading stage, determined via qRT-PCR. The values of qRT-PCR are the mean ± SD (one-way ANOVA with Tukey's test, *n* = 3 biologically independent experiments). **e** qRT-PCR analysis of *Pm13* expression in one-week-old TA3575 seedings at 0, 6, 12, 18, 24, 36, 48, and 72 hpi with the *Bgt* isolate E09. The expression of the genes was evaluated via qRT-PCR, with *TaActin* serving as an endogenous control, and the expression was calculated using the comparative CT method. The values of qRT-PCR are the mean ± SD (one-way ANOVA with Tukey's test, *n* = 3 biologically independent experiments). Source data are provided as a Source Data file.

## Expression pattern of *Pm13*

We visualized the resistance phenotype of *Pm13* with leaves infected with the *Bgt* isolate E09 at 7 d post-inoculation (dpi). TA3575 was highly resistant to *Bgt* isolate E09, with no visible conidia produced (IT 0), while CS was highly susceptible, with a large number of visible conidia (IT 4). DAB and Trypan blue staining showed higher intracellular reactive oxygen species (intraROS) levels and greater cell death in TA3575 compared to CS (Fig. 5a–c). Quantitative RT-PCR (qRT-PCR) analysis revealed that the expression of *Pm13* was highest in flag leaves (Fig. 5d), with expression upregulated with a slight increase at 6 h post-inoculation (hpi), peaking at 12–18 hpi, and decreasing thereafter (Fig. 5e).

To evaluate the impact of *Pm13* on the expression of pathogenesis-related (*PR*) genes involved in plant defense, we assessed the transcript levels of six *PR* genes (*PR1*, *PR2*, *PR3*, *PR4*, *PR5*, and *PR9*) in *Pm13* transgenic lines and the susceptible control Fielder at different time points after inoculation with the *Bgt* isolate E09. Prior to infection, both the *Pm13* transgenic lines and the susceptible control Fielder presented relatively low levels of *PR* gene transcripts. After inoculation, the transcript levels of the *PR* genes increased in both the *Pm13* transgenic lines and the susceptible control Fielder, however, the *Pm13* transgenic lines exhibited significantly greater transcript levels of the *PR* genes than did the susceptible control Fielder (Supplementary Fig. 13). This indicated that fungal effectors probably suppressed the expression of the *PR* genes in the susceptible control Fielder.

## Development of diagnostic markers and desirable introgression line for *Pm13*

The *Pm13* sequence was used to develop two functional markers, AelMLKL-1 and AelMLKL-8 (Supplementary Data 6), that were uniquely

amplified in *Pm13* stocks but were absent in 180 wheat lines lacking *Pm13* (Supplementary Data 15, Supplementary Fig. 14). Furthermore, we developed a T3Sˡ#2S-3BS.3BL recombinant W12-3 (Supplementary Data 4), which has a small 3Sˡ#2 segment harboring *Pm13* in wheat background, the *Ae. longissima* 3Sˡ chromosome segment length in the recombinant wheat chromosome was estimated to be approximately 2.82 Mb based on the *Ae. longissima* TL05 reference genome, which will likely minimize linkage drag. The recombinant W12-3 and diagnostic markers of *Pm13* will facilitate effective deployment of *Pm13* in elite wheat varieties across the world.

## Discussion

Over the past 100 years, the bread wheat gene pool has been endowed with 198 exotic resistance (R) genes, constituting 42% of the total 467 R genes designated in wheat[46]. Approximately 81 Triticeae resistance genes have been cloned to date, of which the common classes encoded NLRs[47]. Recently, resistance genes encoding KFPs have expanded the repertoire of non-NLR genes in the Triticeae tribe for resistance breeding[48]. Here, we report the map-based cloning of the wheat powdery mildew resistance gene *Pm13* and confirm that *AelMLKL* is the causal gene by gain-of-resistance via gene transfer to a susceptible hexaploid wheat variety and suppression and loss-of-resistance through VIGS and EMS mutants. *Pm13* encodes a KFP that contains a kinase domain in its C-terminus fused to the MLKL_NTD domain in the N-terminus, an unusual domain architecture compared to other cloned KFPs (such as *Rpg1*[49], *Yr15*[50], *Sr60*[51], *Sr62*[52], *Pm24*[4], *WTK4*[6], *Rwt4*[53], *Yr36*[54], *Pm4*[5], *Snn3*[55], *Tsn1*[56], *Rpg5*[57], *Sm1*[58], *Lr9*[59], and *Sr43*[47]). KFPs have one apparent functional kinase domain that is fused to a second typically non-functional kinase domain or an entirely different

domain. The integrated domains in KFPs were hypothesized to perceive pathogen effectors, while the kinase catalyzes the phosphorylation of either the effector or the NLR guard to initiate downstream defense responses and immunity[47,48,59]. Evidence for this model is provided by studies of plant NLRs with integrated domains[48,60,61]. Unlike the C-terminal pseudokinase domain of animal MLKLs and plant AtMLKLs, the C-terminal kinase domain of Pm13 comprises conserved residues crucial for kinase activity, as evidenced by sequence alignment[43]. The resistance loss induced by nine mutant sites from eight mutants located in the kinase domain of Pm13 suggested that the kinase domain is essential for Pm13-mediated activation of immune responses (Fig. 2a, b). Nonetheless, whether the C-terminal kinase domain of Pm13 can trigger downstream defense responses and immune reactions has yet to be proven.

The Pm13 domain architecture resembled that of mammalian MLKL, which has an N-terminal MLKL_NTD domain and a C-terminal STK domain. The N-terminal MLKL_NTD domain of Pm13 is strikingly similar to the cell death-inducing HeLo and $CC_R$ domains in animals, plants, and fungi[38,40–42,62]. The N-terminus HeLo domain-containing MLKL protein can mediate necroptosis in animals[38]. In plants, the AtMLKLs HeLo domain can elicit cell death[40]. The $CC_R$ domain present in the RPW8 and ADR1 proteins from *A. thaliana* and the NRG1 protein from *N. benthamiana*, structurally resemble the HeLo domain and have been proven to trigger cell death by disrupting membrane integrity[41,42,63,64]. In fungi, the HeLo domain of HET-S from *Podospora anserina* was also reported to be a membrane-targeting cell death-inducing domain[62,65]. Regulated cell death is intimately connected with innate immunity[40]. Recent studies revealed that a notable similarity of several proteins involved in regulating cell death in different kingdoms is the HeLo domain[38,40,42,62]. Trypan blue staining of the leaves of the Pm13 line TA3575 post *Bgt* isolate E09 infection revealed strong cell death (Fig. 5a, c), which was in accordance with the findings of Li et al.[66]. It is proposed that the N-terminal MLKL_NTD domain of Pm13 may induce cell death and trigger immune signaling. However, it is paradoxical that the functions of the integrated domains fused to kinases in KFPs are presumed to involve the perception of pathogen effectors. Therefore, it is necessary to carry out experiments to determine whether the N-terminal MLKL_NTD domain of Pm13, a member of KFPs family, functions in regulating cell death like other MLKLs or perceiving pathogen effectors like other KFPs.

*Ae. longissima* is a wild diploid species of common wheat that harbors considerable genetic diversity for wheat improvement. To data, several disease resistance genes, including *Pm13*, *Pm66*, and *Pm6Sl*, have been introgressed from *Ae. longissima* into wheat through the development of wheat-*Ae. longissima* recombinants[29–32]. However, no genes for resistance to any biotic or abiotic stresses have been cloned from *Ae. longissima*, which hinders the elucidation of the molecular mechanisms underlying these genes. Recently, the emergence of technologies based on the integration of sequencing and mutagenesis has facilitated the cloning of disease-resistance genes in wheat and its wild relatives. Mutagenesis combined with NLR gene sequencing (MutRenSeq) or chromosome sequencing (MutChromSeq) allowed rapid cloning of the resistance genes *Sr22*[67], *Sr45*[67], *Sr26*[68], *Sr61*[68], *Yr5*[69], *Yr7*[69], *Pm2*[11], *Rph1*[70], and *Sr43*[47]. Mutagenesis combined with mapping and RNA-Seq (MutRNA-Seq) or isoform sequencing (MutIsoSeq) facilitated the cloning of *Sr62*[52], *Lr9/Lr58*[59], *YrNAM*[71] and *Lr47*[72]. Long-read genome sequencing facilitated the cloning of *Yr27*[73] and *Pm69*[8]. Here, we report the cloning of the powdery mildew resistance gene *Pm13* using a combination of *ph1b*-induced recombination and transcriptome sequencing of its direct donor parent TA3575. As a comparison, fine-mapping disease resistance genes such as *Pm13*, which have been introgressed into common wheat from distant wheat wild relatives such as *Ae. longissima*, is relatively tedious due to the low exchange and recombination suppression between the alien chromatin and wheat homoeologous counterpart, which is

strictly controlled by a complex *Ph* system in hexaploid wheat genetic backgrounds. Nonetheless, the deletion mutant of the *Ph1* locus (*ph1b*) at CS 5BL can significantly increase meiotic homoeologous recombination[74] and thus has been widely used to induce wheat-alien homoeologous recombination. Homoeologous recombination induced by *ph1b* is beneficial not only for the map-based cloning of resistance genes that were introgressed into common wheat from its wild relatives but also for the development of introgression lines with shortened alien segments harboring resistance genes for direct use in breeding programs. In this study, a total of 111 wheat-*Ae. longissima* 3S$^l$#2 recombinants were developed based on *ph1b*-induced homoeologous recombination and were genotyped and phenotyped to physically fine-map *Pm13* to a 124.43 kb genomic region on the *Ae. longissima* TL05 reference sequence. Moreover, a resistant introgression line harboring *Pm13* with a 2.82 Mb 3S$^l$#2 genomic length was developed, providing an opportunity for both cloning and utilizing this resistance gene from a wild relative of wheat.

In the recent past, the availability of annotated genomes of a few wild wheat plants along with the genome sequences of more than 14 hexaploid or tetraploid wheat accessions has greatly facilitated the map-based cloning of disease resistance genes in wheat[36,37,52,75–79]. However, due to large sequence differences among different accessions within the same species, the availability of one reference genome sequence sometimes hinders the ability to identify functional genes in different accessions of the same species or related species[14,16]. In the present study, the *Pm13* genomic sequence from the *Ae. longissima* accession TL20 was not found in the *Ae. longissima* reference genome of accession TL05. To overcome this constraint, we combined physical mapping of *Pm13a* and transcriptome sequencing of its direct donor parent TA3575 to successfully discover that CL897Contig1 is a serine/threonine protein kinase from the TA3575 transcriptome, which was further confirmed to be *Pm13a*. Recently, Zou et al. reported sequence differences around the *Pm60* locus in the donor parent *T. urartu* accession PI428309 corresponding to the reference genome sequence of *T. urartu* accession G1812[14], limiting progress in characterizing of this locus using a reference genome sequence. Under these circumstances, our study showed that transcriptome sequencing of direct donor of target genes may help avoid potential problems caused by genomic differences in the reference genome or incomplete reference genome assemblies. Thus, combining *ph1b*-induced recombination for fine-mapping and RNA-Seq of direct donors of target genes is a powerful tool for isolating resistance genes that have been introgressed into the hexaploid wheat background from wild species.

*Pm13*, located on the short arm of chromosome 3S$^l$#1 (3S$^l$#1S) of the *Ae. longissima* accession TL01 was subsequently transferred into wheat in 1988 via the induction of CS-*Ae. longissima* T3S$^l$#1S-3BS.3BL and T3S$^l$#1S-3DS.3DL recombinants[28]. The gene has maintained its effectiveness against different *Bgt* biotypes for more than 30 years across several countries[29,80]. However, the deployment of this gene has been limited in wheat breeding programs due to linkage drag, which results in inferior agronomic characteristics owing to the presence of a larger chromosomal segment of *Ae. longissima*[81]. In the present study, we developed a resistant T3S$^l$#2S-3BS.3BL recombinant W12-3 strain that carried *Ae. longissima* segments harboring *Pm13* as short as <0.35% of the 3S$^l$#2 genomic length. This resistant introgression line could minimize the undesirable linkages associated with *Pm13*. Furthermore, we designed two functional markers, AelMLKL-1 and AelMLKL-8, from *Pm13* for precise marker-assisted selection of common wheat lines harboring *Pm13*. The W12-3 line and two *Pm13* functional molecular markers will facilitate effective wide-scale deployment of *Pm13* in elite wheat varieties.

In summary, we cloned the wheat powdery mildew resistance gene *Pm13* from *Ae. longissima*. *Pm13* encodes a MLKL protein that contains a MLKL_NTD domain in N-terminal and a STK domain in its C-terminus. Isolation of *Pm13* will shed light on its disease resistance

mechanism. In addition, the resistance introgression line harboring *Pm13* within approximately 2.82 Mb of the 3S$^l$#2 genomic length (estimated using *Ae. longissima* TL05 reference genome) and the developed diagnostic functional markers AelMLKL-1 and AelMLKL-8 for *Pm13* will facilitate deployment of this gene in elite wheat varieties through MAS or by engineering gene pyramids that may maximize its longevity in powdery mildew resistance.

## Methods

### Plant materials
The CS-*Ae. longissima* recombinant lines T3S$^l$#1S-3BS.3BL (R1B) and T3S$^l$#1S-3DS.3DL (R2B) harboring *Pm13* were developed by Ceoloni et al.[28] and provided by Nanjing Agricultural University, China. The CS-*Ae. longissima* 3S$^l$#2(3B) disomic substitution line TA3575, the CS-*Ae. longissima* 3S$^l$#3S isochromosome addition line TA7545, the wheat landrace CS, the CS *ph1b* mutant TA3809 which lacked the *Ph1* gene and thereby elevated homoeologous recombination, and the *Ae. longissima* accession TA1910 (TAM4) were kindly provided by the Wheat Genetics Resource Center at Kansas State University, USA, and maintained at the Experimental Station of Henan Agricultural University, China. The donors of chromosomes 3S$^l$#1 in R1B, 3S$^l$#2 in TA3575, and 3S$^l$#3 in TA7545 were derived accordingly from the *Ae. longissima* accessions TL01, TL20 and TA1910 (TAM4)[82–84]. The powdery mildew-susceptible wheat cultivar Fielder was used for the transformation of *Pm13*. One hundred and eighty wheat lines (Supplementary Data 15) were used for verifying diagnostic markers for *Pm13*. The plants were grown in a greenhouse with 14 h light/10 h dark (22/18 °C, 70% relative humidity).

### Powdery mildew resistance phenotyping
*Bgt* isolate E09 and another 35 isolates with different virulent spectrums from different provinces of China (Supplementary Data 7) were used to evaluate powdery mildew resistance under controlled conditions. *Bgt* isolates were maintained and increased on the highly susceptible CS seedlings. Wheat seedlings at the first-leaf stage (one-week-old) were inoculated with *Bgt* isolates and then grew in a greenhouse at 22 °C under a 14 h light/10 h darkness photoperiod[30]. Ten individual plants for each of R1B, TA3575, TA7545, and CS were inoculated with 36 *Bgt* isolates. At least 16 individuals of F$_2$ progenies from each 3S$^l$#2 recombinant, M$_2$, and M$_3$ families of EMS-treated mutants and transgenic families were inoculated with *Bgt* isolate E09 to evaluate resistance phenotypes. Infection types (ITs) were assessed at 7 dpi using a scale from 0 to 4[85]. The phenotypes were divided into two categories, resistant (R, IT 0-2) and susceptible (S, IT 3-4). To visualize the fungal structures, the first leaves of TA3575 and CS were cut into 2 cm segments at 7 dpi with *Bgt* isolate E09 and fixed in a fixative (ethanol:glacial acetic acid, 3:1, v/v) for 12–16 h. Then tissue was rinsed with deionized water twice and stained with Coomassie brilliant blue (0.05% Coomassie brilliant blue R250, 10% acetic acid, 50% methanol) for 10–15 min[86]. The treated leaves were observed under the Olympus BX51 microscope (Olympus Corporation, Tokyo, Japan).

### Detection of ROS accumulation and plant cell death
To detect the ROS accumulation, the first leaves were cut from seedlings of TA3575 and CS at 48 hpi with *Bgt* isolate E09 and immediately incubated in a 3,3′-diaminobenzidine (DAB) solution (1 mg/mL, pH 5.8) for 12 h, and then bleached in destaining solution (ethanol:acetic acid, 1:1, v/v) for 12–16 h. The bleached leaves were stained with 0.6% (w/v) Coomassie blue solution for 10 s and then washed with water for assessing the ROS accumulation[87]. To detect plant cell death, the first leaves from TA3575 and CS plants at 48 hpi with *Bgt* isolate E09 were incubated in a 0.4% Trypan blue solution for 2 min in boiling water, then bleached in chloral hydrate solution (2.5:1, w/v) for 24 h. The bleached leaves were incubated in 0.6% (w/v) Coomassie blue solution

for 10 s[88]. ROS-induced epidermal cell death was observed under an Olympus BX51 microscope (Olympus Corporation, Tokyo, Japan). All microscopic experiments were conducted with three biological replicates, and at least 100 infected cells were observed for each replicate to determine ROS accumulation and cell death responses. The quantitative assessment of ROS accumulation and associated cell death was performed[66].

### Initial mapping of *Pm13a* using CS *ph1b* mutant
To develop 3S$^l$#2 recombinants, the CS-*Ae. longissima* 3S$^l$#2(3B) disomic substitution line TA3575 was crossed with the CS *ph1b* mutant TA3809. The derived BC$_1$F$_1$ populations were screened to identify individuals homozygous for *ph1b* and monosomic for 3S$^l$#2 using *ph1b*-specific marker ABC302.3 designed by Wang et al.[35] and 3S$^l$#2-specific markers developed in this study. The 1580 selected individuals-derived BC$_1$F$_2$ progenies were analyzed by four 3S$^l$#2-specific markers (CL54397, CL1543, CL90315, and CL88266) in the distal and proximal regions of each arm of chromosome 3S$^l$#2 to select CS-*Ae. longissima* 3S$^l$#2 recombinants missing one to three markers. The recombinants were further characterized using other 3S$^l$#2-specific molecular makers. The recombinants and derived progenies were phenotyped for resistance to the *Bgt* isolate E09 for initial mapping of *Pm13a*. All information for the 3S$^l$#2-specific markers is listed in Supplementary Data 1. PCR was performed in 15.0 μL volumes containing 2.0 μL template gDNA (100 ng/μL), 1.0 μL of each primer (5.0 μmol/L), 7.5 μL 2 × Es Taq MasterMix (CWBIO, Beijing, China) and 3.5 μL ddH$_2$O. The PCR program consisted of initial denaturation at 95 °C for 5 min followed by 35 cycles of 95 °C for 30 s, 53–66 °C annealing for 30 s depending on different primers, and 72 °C for 1 min, with a final extension at 72 °C for 10 min. To obtain sequence polymorphisms, the PCR products without polymorphism were digested with restriction enzymes *Mse*I, *Msp*I, *Mbo*I, *Alu*I, and *Rsa*I (New England Biolabs, Beijing, China). Five microlitres of a restriction enzyme mixture containing 2.85 μL of ddH$_2$O, 2.0 μL of CutSmart buffer, and 0.15 μL of an enzyme stock solution was added to 15.0 μL of PCR products and incubated for 3.5 h at 37 °C. The PCR or restricted PCR products were separated via gel-electrophoresis on a 2.0% agarose gel stained with ethidium bromide and visualized by UV light.

### High-resolution mapping of *Pm13a*
For fine mapping of *Pm13a*, 8530 F$_2$ individuals derived from heterozygous 3S$^l$#2 resistant recombinants were phenotyped for powdery mildew resistance and genotyped with *Pm13a*-flanking markers CL10208 and CL56281 to develop 3S$^l$#2 recombinant population for high-resolution mapping in the *Pm13a* interval. The *Ae. longissima* genome sequence[36,37] was used to design additional markers in the mapping interval. The markers that cosegregated with *Pm13a* in the initial map interval were also subjected to fine-mapping. The *Pm13a* gene was physically fine-mapped by combining genotypic and phenotypic data for all the recombinants.

### RNA-Seq of *Pm13a* donor parent TA3575
Transcriptome sequencing of the *Pm13a* donor parent TA3575 and recipient parent CS was described as Dong et al.[34]. In brief, TA3575 and CS were inoculated with the *Bgt* isolate E09 when the first leaves were fully unfold. Leaves at 0, 12, 24, 48, and 72 hpi were respectively collected and immediately frozen in liquid nitrogen for RNA extraction. Total RNA was extracted from ten samples (0, 12, 24, 48, and 72 hpi for TA3575 and CS, respectively) for RNA-seq at OE Biotech (Shanghai, China). Then, equal amounts of RNA from TA3575 and CS at 12–72 hpi were mixed to generate RNA-seq samples TA3575_I and CS_I, respectively. RNA at 0 hpi from TA3575 and CS was accordingly represented as RNA-seq samples TA3575_C and CS_C. Two biological replicates were performed, forming a total of eight RNA samples (TA3575_I1, TA3575_I2, TA3575_C1, TA3575_C2, CS_I1, CS_I2, CS_C1, and CS_C2).

Designations 1 and 2 are used to represent replicates 1 and 2, respectively.

## Cloning of full-length CDS of AelMLKL gene

Total RNA was extracted from the E09-inoculated leaf tissue of the resistant line TA3575 using RNA isolater Total RNA Extraction Reagent (Vazyme, Nanjing, China). Reverse transcription was performed using HiScript III 1st Strand cDNA Synthesis Kit (+gDNA wiper) (Vazyme, Nanjing, China) following the manufacturer's instructions. The full-length CDS of the *AelMLKL* gene was amplified using 2 × Phanta Flash MasterMix (Dye Plus) (Vazyme, Nanjing, China) by PCR with the gene-specific primer AelMLKL_CDS (Supplementary Data 6). The PCR product was cloned into the One step ZTOPO-Blunt/TA vector (ZOMAN-BIO, Beijing, China). The inserted fragment in the ZTOPO-Blunt/TA-AelMLKL_CDS construct was verified by Sanger sequencing at Sangon Biotech, Shanghai, China.

## EMS mutagenesis and mutant validation

Seeds of TA3575 were treated with EMS as described by Ma et al.[89]. Specifically, a total of 4800 seeds were soaked in distilled water for 6 h and treated with 0.6% EMS solution on a shaker at 150 rpm for 14 h. $M_0$ seeds were sown in the field with 80 seeds per 2 meter-length row, and 506 $M_1$ plants were harvested. Sixteen $M_2$ seedlings from each $M_1$ line were inoculated with *Bgt* isolate E09 under controlled greenhouse conditions to identify susceptible mutants. The $M_3$ progenies derived from susceptible $M_2$ mutants were challenged again with *Bgt* isolate E09 to confirm their susceptibility and select homozygous mutants. The $M_3$ mutant lines were further validated by 3S$^l$#2-specific markers, genomic in situ hybridization (GISH), and fluorescence in situ hybridization (FISH) analyses. The gDNA and full-length CDS of the *AelMLKL* gene were amplified from each of the $M_3$ homozygous mutants using the primers listed in Supplementary Data 6. The PCR products were sequenced at Sangon Biotech, Shanghai, China. The *AelMLKL* gene sequences from the mutants and wild-type TA3575 were compared using DNAMAN 8 software (Lynnon Biosoft, San Ramon, CA, USA) to identify SNPs.

## Cytogenetic analyses

GISH and FISH was used to rule out EMS-induced $M_2$ susceptible plants caused by seed contamination or loss-of 3S$^l$#2 chromosomes and verify loss-of-function mutants. Chromosome spreads of root tip cells at mitotic metaphase were prepared following Huang et al.[90]. Root tips were exposed in an $N_2O$ gas chamber for 2 h followed by a 30-min treatment with 90% acetic acid. Subsequently, the root tips were rinsed three times with 75% ethanol and fixed in a fixative (ethanol:glacial acetic acid, 3:1, v/v) for 2–7 days. For GISH, total gDNA of *Ae. longissima* accession TL20 was labeled with fluorescein-12-dUTP and used as the probe, and unlabeled gDNA of common wheat CS was served as blocking[30]. The ratio of gDNA of *Ae. longissima* to CS was 1:130. After GISH, the hybridization signals were washed with phosphate-bufered saline (PBS) to conduct dual-color FISH using eight single-strand oligonucleotides pSc119.2-1, (GAA)$_{10}$, pAs1-1, pAs1-3, pAs1-4, pAs1-6, AFA-3 and AFA-4 as probes[90,91]. The first two oligonucleotides were modified with 6-carboxyfluorescein (FAM), and the last six oligonucleotides were modified with 6-carboxytetramethylrhodamine (TAMRA) at the 5′-ends. Fluorescent images were captured with AxioCam MRc5 CCD camera using a Zeiss Axio Scope A1 fuorescence microscope (Germany). Images were processed using Photoshop CS 3.0 (Version 10.0.1). The oligo probes were synthesized with 5′-ends labeled with TAMRA or FAM by Sangon Biotech, Shanghai, China.

## Virus-induced gene silencing

BSMV-VIGS was used to investigate the functions of the *AelMLKL* gene in TA3575 as described by Xing et al.[12]. In brief, a 246-bp target fragment with little homology (identity <50%, stretches of 100% nucleotide identity <21-nt) in the wheat assemblies was inversely inserted into the γ-strain of BSMV by replacing the *TaPDS* sequence of the BSMV:*TaPDSas* vector for construction of the BSMV:*AelMLKL as* recombinant silencing vector. Plasmid linearization, in vitro transcription, and virus inoculation were performed as described by Wang et al.[92]. In particular, the first fully expanded leaves of TA3575 seedlings were infected with the in vitro-transcribed BSMV:*AelMLKL as* virus, BSMV:*TaPDSas* and BSMV:γ served as controls. At 14 d after virus infection, when viral infection symptoms were clearly visible, the third leaves were detached and placed on 1% agar plates supplemented with 20 mg/mL 6-phenyladenine (6-BA). Then, the detached leaves were infected with the fresh *Bgt* isolate E09 and incubated at 24 °C under a 14-h light /10-h dark photoperiod with 70% humidity. After 7 days inoculation, powdery mildew disease phenotypes were evaluated and leaf samples were collected for further gene silencing expression analyses of *AelMLKL*. At least 15 plants were challenged by BSMV vector, and the experiments were repeated three times.

## Plant genetic transformation

The CDS of *AelMLKL* flanked by a *Bam*HI restriction site was amplified via PCR from the ZTOPO-Blunt/TA-AelMLKL_CDS construct using the primer set AelMLKL_OETrans and cloned into the *Bam*HI site of the pWMB110 vector downstream of the maize (*Zea mays* L.) ubiquitin (*Ubi*) promoter with the nopaline synthase (*Nor*) terminator via the ClonExpress II One Step Cloning Kit (Vazyme, Nanjing, China) to generate the overexpression vector *ProUbi*:*AelMLKL*. The colonies were sequenced to confirm the accuracy of the cDNA insertions. The *ProUbi*:*AelMLKL* plasmid was transformed into the *Agrobacterium tumefaciens* strain EHA105 and transformed into the susceptible wheat cultivar Fielder. Specific marker AelMLKL_OE was used to detect the presence of *AelMLKL* genes in the transgenic progenies. PCR primer pair AelMLKL_Exp was used to detect the expression of *AelMLKL* in the transgenic plants. All the primers used are listed in Supplementary Data 6. Sixteen $T_1$ transgenic plants (first-leaf stage) for each transgenic event were tested for their responses to *Bgt* isolate E09, and disease symptoms were recorded at 7 dpi.

## Gene expression analysis

Total RNA was extracted from the first leaves of the *Pm13* transgenic lines AelMLKL-OE11 and AelMLKL-OE14, and susceptible control Fielder at the first-leaf stage before inoculation (0 hpi) and 12, 18, 24, 36, and 48 hpi with *Bgt* isolate E09 using TRIzol reagent (Vazyme, Nanjing, China). Each time point was composed of three biological replicates with five individuals per replicate. Reverse transcription was performed using HiScript III RT SuperMix for qPCR (+gDNA wiper) (Vazyme, Nanjing, China). qRT-PCR analysis was carried out using SYBR Mix (TaKaRa, Dalian, China) following the manufacturer's instructions. The primers used to evaluate the transcript levels of the gene *AelMLKL* and *PRs* are listed in Supplementary Data 6. The wheat *TaActin* gene was used as the endogenous control[4]. The comparative CT method was used to quantify relative gene expression[93]. For validation of VIGS, three biological replicates with five leaves per replicate were used for expression analysis.

## Allelism at the *Pm13* locus

To test the allelism of *Pm13*, *Pm13a*, and *Pm13b*, reciprocal crosses were made with R1B (*Pm13*), TA3575 (*Pm13a*), and TA7545 (*Pm13b*). All the F$_2$ plants derived from the crosses R1B × TA3575 (2183 plants), R1B × TA7545 (2715 plants), TA3575 × R1B (2785 plants), TA3575 × TA7545 (2176 plants), TA7545 × R1B (2082 plants) and TA7545 × TA3575 (2,168 plants) were phenotyped by inoculation of *Bgt* isolate E09 to investigate the segregation for powdery mildew resistance. Furthermore, the resistance spectrum of R1B, TA3575, and TA7545 were also assayed using 36 *Bgt* isolates with different virulent spectrum collected from the main wheat-growing provinces of China. To further

test whether the *Pm13*, *Pm13a*, and *Pm13b* genes are identical, the 6552 bp genomic sequences of *AelMLKL* gene were obtained from R1B (*Pm13*) and TA7545 (*Pm13b*) using the primers listed in Supplementary Data 6. The 15 μL DNA amplification system contained 7.5 μL 2 × Es Taq MasterMix (CWBIO, Beijing, China), 2.0 μL template gDNA (100 ng/μL), 1.0 μL of each primer (5.0 μmol/L) and 3.5 μL ddH2O. DNA amplification was performed at 95 °C for 5 min followed by 35 cycles of 95 °C for 30 s, 50–60 °C (depending on specific primers) for 30 s, and 72 °C for 1 min, with a final extension at 72 °C for 10 min. The PCR products were sequenced by Sanger dye-terminator method at Sangon Biotech, Shanghai, China. The sequences of the *AelMLKL* gene from R1B (*Pm13*) and TA7545 (*Pm13b*) were compared with that from TA3575 using DNAMAN 8 software (Lynnon Biosoft, San Ramon, CA, USA).

### Subcellular localization analysis

To determine the subcellular location of Pm13, the *Pm13* coding sequence was cloned and inserted into the pJIT163-GFP vector between *Hin*dIII and *Eco*RI sites for *Pm13-GFP* fusion expression driven by the *CaMV35S* promoter (*35S::Pm13-GFP*) in wheat protoplasts. The *35S::Pm13-GFP* was cotransfected into protoplasts with the nucleus marker plasmid *AtPIF4-mCherry* using polyethyleneglycol (PEG)-calcium-mediated transfection[94,95]. The transformed protoplasts were cultured at 25 °C for 16 h under dark conditions, and observed using a laser scanning confocal microscope (A1HD25Nikon, Tokyo, Japan). Image acquisition was conducted with NIS-Elements Viewer Imaging Software (version 5.21.00). The subcellular localization of Pm13 protein was determined for three times.

### Protein sequence and domain analysis

Prediction of the core domain of AelMLKL was performed based on the conserved domain database from the NCBI. Three-dimensional structure modeling of AelMLKL was performed using the open source code of AlphaFold v2.0[39] with the amino acid sequence of AelMLKL as input. The output consisted of five ranked models in.pdb format. We utilized the rank_1.pdb model that contains the best predicted local distance difference test (pLDDT) score at 78.8. The 3D structural graphics were generated using PyMOL (v.2.6.0a0). Substituted residues that did not exhibit any visual surface on the 3D model were characterized as internal residues following Wang et al.[59].

### Phylogenetic analysis of Pm13

We first retrieved all MLKL_NTD domain-containing (CDD cd21037) proteins and DUF1221 domain-containing proteins from *Poaceae* and *Arabidopsis* in the Interpro database (https://www.ebi.ac.uk/interpro). Subsequently we performed a BLASTP analysis with the MLKL_NTD domain of Pm13 across the whole-genomes of *T. aestivum* (cv. Chinese Spring), *T. urartu* (cv. G1812), *Ae. tauschii* (cv. AL8/78), *T. turgidum* dicoccoides (cv. Zavitan), *T. turgidum* durum (cv. Svevo), *Hordeum vulgare* (cv. Morex), *Secale cereale* (cv. Weining), *Thinopyrum elongatum* (cv. D-3458), and five Sitopsis species of *Aegilops* genus (*Ae. bicornis* cv. TB01, *Ae. longissima* cv. TL05, *Ae. searsii* cv. TE01, *Ae. sharonensis* cv. TH02, and *Ae. speltoides* cv. TS01) at WheatOmics (http://wheatomics.sdau.edu.cn/)[96] with the default setting. The MLKL_NTD domains of the corresponding proteins in the output from WheatOmics were verified in the NCBI Conserved Domain Search (https://www.ncbi.nlm.nih.gov/Structure/cdd/wrpsb.cgi). The phylogenetic trees were constructed using the neighbor-joining method in MEGA7[97] and drawn with iTOL (https://itol.embl.de/). Collinearity analysis among different species or subgenomes was performed using the online tool Triticeae-GeneTribe with the default parameters[98].

### Validation of *Pm13* functional molecular markers

CS, CS-*Ae. longissima* 3SˡŁ#2(3B) disomic substitution line TA3575, CS-*Ae. longissima* T3SˡŁ#1S-3BS.3BL recombinant R1B, CS-*Ae. longissima* T3SˡŁ#1S-3DS.3DL recombinant R2B, CS-*Ae. longissima* T3SˡŁ#2S-3BS.3BL recombinant W12-3 with tiny 3SˡŁ#2 segment harboring *Pm13* developed in this study and 180 wheat lines (Supplementary Data 15) were used to validate *Pm13* functional molecular markers AelMLKL-1 and AelMLKL-8. Genomic DNAs of leaf tissues from these plant materials were extracted using cetyltrimethylammonium bromide (CTAB) method. PCRs were performed in 15 μL volumes containing 7.5 μL 2 × Es Taq MasterMix (CWBIO, Beijing, China), 1.0 μL of each primer (5.0 μmol/L), 2.0 μL template gDNA (100 ng/μL) and 3.5 μL ddH2O. The PCR thermocycling conditions were initial denature at 95 °C for 5 min, 35 cycles of 95 °C for 30 s, 58 °C for 30 s, and 72 °C for 1 min, followed by a final extension at 72 °C for 10 min. PCR products were separated via gel-electrophoresis on a 2.0% agarose gel stained with ethidium bromide and visualized by UV light.

### Reporting summary

Further information on research design is available in the Nature Portfolio Reporting Summary linked to this article.

## Data availability

Data supporting the findings of this work are available within the paper and its Supplementary Information files. The plant materials and datasets generated and analyzed during the present study are available from the corresponding authors upon request. Detailed sequence data of *Pm13* gene was deposited in NCBI database under accession OQ784172. Public databases WheatOmics (http://wheatomics.sdau.edu.cn/) and Interpro database (https://www.ebi.ac.uk/interpro) were used for this study. Source data are provided with this paper.

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

## Acknowledgements

We are grateful to Prof. Bikram Gill from Wheat Genetic and Genomic Resources Center, Department of Plant Pathology, Throckmorton Plant Sciences Center, Kansas State University, Manhattan, USA, for

critically reading and improving the manuscript. This work was financially supported by the National Natural Science Foundation of China (32272070 to H.H.L., 32372089 to W.X.L. and 31971887 to W.X.L.), the Scientific and Technological Research Project of Henan Province of China (212102110059 to W.X.L.), the Topnotch Talents of Henan Agricultural University (30500939 to H.H.L.) and South Dakota Wheat Commission and USDA AFRI 2022-68013-36439 (WheatCAP) to S.K.S.

## Author contributions

W.X.L., H.H.L., and S.K.S. designed the study. H.H.L., W.Q.M., C.M., Z.J.D., X.B.T., C.L.W., P.T.M. and H.S.G. performed the research. H.H.L., W.X.L., W.Q.M., Y.Z., S.K.S., Q.W.L., C.L., Z.B.Z. and B.L. analyzed the data. H.H.L., W.X.L., and S.K.S. wrote the paper and all authors contributed to revision and editing.

## Competing interests

The authors declare no competing interests.
