## [Peer Review File · Nature Communications]

Wheat powdery mildew resistance gene Pm13 encodes a unique mixed lineage kinase domain-like proteinReviewers' Comments:

Reviewer #1:

Remarks to the Author:

Li et al. present their work on cloning the wheat powdery mildew resistance gene Pm13 that is derived from *Aegilops longissima*. Several approaches were integrated to identify a mixed lineage kinase-like (MLKL) protein as the causal gene including a forward genetic screen using mutagenesis, transient virus induced gene silencing, and transgenic complementation. Comparison of Pm13 with other known MLKL family members found structural similarity despite substantial sequence divergence. As part of this work, the authors develop a small introgression of Pm13 and provide breeders markers for the deployment of this gene in elite cultivars.

Major concern 1. The identification of an MLKL encoding gene as causal for disease resistance has not been described to date. Previous work by Mahdi et al. (2020) identified the role of MLKL proteins in plant immunity and cell death processes, but their work was not based on natural genetic variation and only identified a small number of proteins described as MLKLs (typically 2 to 3 per species). A preliminary bioinformatic analysis of AtMLKL1, AtMLKL2, and AtMLKL3 found that the N-terminal domain is classified as a DUF1221, whereas Pm13 is classified as carrying an MLKL N-terminal domain (CDD cd21037). While the authors have used AlphaFold2 to infer the potential structural relationship of Pm13 and AtMLKL1, there lacks a complete analysis of the presence of this domain structure in other plant species. The CDD database (<https://www.ebi.ac.uk/interpro/entry/cdd/CD21037/>) suggests that over 1k proteins carry the cd21037 domain in grasses, indicating that this overall family may be larger than previously described. The current work does not sufficiently place the impact of the discovery within the context of gene family evolution (specifically, MLKL_NTD-PK). This work has several implications:

A. It appears that the original finding by Mahdi et al. underestimated the number of MLKLs in plants. Therefore, the authors should characterize this extended set of proteins that may belong to the MLKL family.

B. The existing phylogenetic tree built for the protein kinase domain is insufficient. No bootstrapping has been performed and the domains themselves have not been placed within the overall context of protein kinase domains in angiosperms. See "Diversity, classification and function of the plant protein kinase superfamily" by Melissa D. Lehti-Shiu and Shin-Han Shiu for a protocol on classifying the protein kinase domain in Pm13.

C. It was unclear which data sets that the authors have investigated to identify the presence of Pm13 in the Triticeae. I suggest a Supplemental Table that documents the presence or absence in diverse data sets analyzed. I noted that Pm13 appears to be naturally varying in *Aegilops bicornis*, *Aegilops longissima*, and *Aegilops sharonensis*. I suggest the authors investigate publicly available RNAseq data sets in NCBI.

Major concern 2. The authors specifically state that the protein kinase domain of Pm13 is functional but they provide no molecular evidence for this claim (such as lines 200-202). The text either needs to sufficiently temper this statement (such as "Pm13 is predicted to be a functional protein kinase based on the presence of key residues for kinase activity") or kinase assays need to be performed. Regardless, the authors need to prepare a supplementary figure where they show the position of the conserved residues for kinase activity.

Major concern 3. A statement about broad-spectrum resistance has to be tempered with the diversity of isolates used for assessment. In this work, the isolates are predominantly only from China (although I believe reference 29 tested a single Italian isolate), therefore I suggest the authors temper their statements and let the passage of time show the utility of Pm13.

Minor concerns

1. Throughout, the references appear to be incorrect in several places.

2. Throughout, CS was not defined as Chinese Spring.
3. Abstract, "due to linkage drag" to "due to linkage drag with deleterious traits".
4. Abstract, "The new lines..." to "Our sequential reduction of the genomic segment encompassing Pm13 will facilitate wide-scale deployment in wheat".
5. Introduction, lines 48-49, the language of the sentence is unclear. Perhaps consider "Recent work has shown that the Triticeae contain an extended array of resistance genes that include diverse kinase domains." I would avoid associating the presence of a kinase domain with durability, as this has not been shown in any conclusive systematic manner.
6. Introduction, line 73, "at chromosome" to "located on chromosome" also "at 6S" to "located on 6S".
7. Introduction, line 80, "by a strategy" to "using a strategy"
8. Results, line 91, "In absence" to "In the absence"
9. Results, line 92, "reference genome we performed transcriptome sequences to" to "reference genome, we performed transcriptome sequencing to"
10. Results, line 124, "Thus, gene" to "Thus, the gene"
11. Results, lines 133 to 139. The authors identify 14 loss-of-function mutants but only provided discussion on the 11 loss-of-function mutants in Pm13. The authors do not comment on complementation tests (Pm13 and non-Pm13 mutants), nor the potential role of these mutants in elucidating Pm13 function.
12. Discussion, line 315. Change "However, the gene was deployed only in a few wheat..." to "However, the gene had limited deployment in wheat..."
13. Discussion, lines 319-320. I suggest tempering this sentence, as the authors do not show if this line does not suffer deleterious effects in the field. I agree that the line is promising.
14. Discussion, line 330. Change "tiny" to the actual size.

Reviewer #2:

Remarks to the Author:

In this manuscript, Li and colleagues cloned the powdery mildew resistance gene Pm13 that was introgressed into bread wheat from the wild wheat relative *Aegilops longissima*. The authors use genetic mapping, mutagenesis, gene silencing and transgenic complementation and show that Pm13 encodes an unusual kinase fusion protein. The manuscript adds to the growing number of kinase fusion proteins involved in disease resistance in Triticeae. The study is well executed and of interest to the broad readership of Nature Communications, contributing novel insights into one of the hottest topics in the field of host-pathogen interactions in cereals (the role of unusual kinase fusion proteins in plant-pathogen interactions). I do have some comments that need to be addressed before publication:

Major comments:

- For the transformation, the authors used an overexpressing construct (Ubi promoter) and the Pm13 coding sequence. There are several examples of disease resistance genes that have possibly been wrongly identified in wheat because of candidate gene overexpression. A critical experiment is to demonstrate that the overexpressing transgenic lines maintain race-specificity. According to literature, virulence to Pm13 has been reported (for example Zhang et al. 2022 PeerJ 10: e14118). Ideally, the authors can test a virulent Bgt isolate on some of their transgenic lines to prove that the resistance seen in the transgenic lines is not caused by a general activation of immunity by the overexpressing the Pm13 CDS.
- The authors state that the HeLo-kinase fusion is unique to *Sitopsis* species of the genus *Aegilops* and absent in bread wheat and other wild wheat relatives. This would indicate a rather recent fusion event. The manuscript could be greatly improved by adding additional information regarding the possible evolution of Pm13. For example, the authors show in Supplementary Fig. 7 that the Pm13 kinase domain is similar to kinase domains of other kinase fusion proteins involved in disease resistance, including Pm4b and WTK2. But what about the N-terminal HeLo domain? What are the closest homologs in bread wheat when only using the HeLo domain? Are HeLo domains usually fused to other

domains in bread wheat and other Triticeae species? This could help to make predictions about the possible function of Pm13. For example, is it possible that the HeLo domain in Pm13 is an integrated decoy?

Minor comments:

- Abstract: 'Thus, Pm13 represents a special class of plant resistance genes revealing a novel aspect of host-pathogen interactions.' Pm13 definitely has a novel and unusual domain architecture, but I would not call it a 'special class' that reveals a 'novel aspect of host-pathogen interactions'. A good number of Triticeae disease resistance genes encode kinase fusion proteins and they probably all follow a similar molecular mechanism. This has recently been discussed (<https://www.researchsquare.com/article/rs-1820134/v1>), resulting in the integrated decoy model. I think that the Pm13 gene might fall into this category of kinase fusion proteins and should be discussed with respect to previous findings (see also my comment above regarding a deeper evolutionary analysis of the two domains).
- Abstract: The authors state that Pm13 has not been widely used in breeding. I would thus refrain from using the term 'broad-spectrum'. In my view, this term should only be used if a disease resistance gene maintains broad-spectrum specificity when exposed to the pathogen in agriculture.
- Line 43: 'can lead to an estimated yield loss of up to 40%'. This is true, but a more realistic estimation of the average yield losses caused by powdery mildew is around 1% (Supplementary Table 3 of Savary et al. 2019 Nature Ecology & Evolution 3: 430-439).
- Lines 48-52: I don't think that there is any evidence that kinase fusion proteins are more durable compared to NLR-mediated immunity. Also, it is not clear if kinase fusion proteins function through a 'different molecular mode' as suggested here. This statement is misleading because it suggests that kinase fusion proteins provide more durable resistance and function through a novel molecular mechanisms, for which there is no evidence yet.
- Line 70-71: What does the nomenclature 3Sl#1S, 3Sl#2S, etc. refer to? Is this the same chromosome from different Ae. longissima accessions?
- Line 91-92: Obviously, an Ae. longissima reference genome was produced during this project, but was not available at the beginning of this project. This can be slightly confusing. The sentence here could be re-phrased to be more clear. For example 'Because no Ae. longissima reference genome was available at the beginning of this project, we used transcriptome sequencing to design a set of 43 3Sl#2-specific markers.'
- Lines 168-169: Please provide the number of populations and the number of F2 plants used for the allelism tests. The numbers are provided in Supplementary Table 9, but it would be beneficial for readers to also have them in the main text.
- Line 175: I suggest writing 'thus providing strong evidence that Pm13, Pm13a, and Pm13b are the same gene (Pm13 hereafter)'.
- Line 186: 'confirming that both domains and connection brace are required...' should be replaced with 'indicating that both domains and connection brace might be required'. It is also possible that the amino acid substitutions affect protein stability and protein abundance as a whole.
- Line 239: 'A wide majority of the R genes cloned till date encode NLRs'. I would argue that this statement is probably no longer true. The number of cloned kinase fusion proteins in Triticeae increased to 17 with the addition of Pm13 and Pm57, which is still less compared to NLR-encoding genes, but NLRs are no longer a 'wide majority' in Triticeae.
- Lines 241-243: Here, the authors imply again that kinase fusion gene stacks might be more durable compared to NLR gene stacks. There is no experimental evidence for this and this statement should be rephrased.
- Lines 248-252: Although Pm13 has a different domain architecture compared to other kinase fusion proteins, it is highly unlikely that Pm13 functions through a 'special disease resistance mechanism'. Two preprints recently suggested a common model for kinase fusion proteins (<https://www.researchsquare.com/article/rs-1820134/v1> and <https://www.researchsquare.com/article/rs-1807889/v1>) named the integrated decoy model. I think it would make more sense to discuss Pm13 in the context of this model, rather than suggesting a new

molecular mechanism (see also my major comment regarding the evolutionary history of Pm13).

Reviewer #3:

Remarks to the Author:

Li et al. reported the map-based cloning of broad-spectrum powdery mildew resistance gene Pm13 from a wild wheat relative *Aegilops longissima*. The cloning method and proofs are solid and convincing. The candidate gene was validated by loss-of-function mutants, gene silencing, transgenic assay, and allelic association analyses. Interestingly, Pm13 encodes a novel mixed lineage kinase domain-like (MLKL) protein which contains an N-terminal HeLo domain and a C-terminal authentic serine/threonine kinase (STK) domain, a unique domain architecture that has not been reported before for plant disease-resistance genes. Cloning of Pm13 will benefit wheat breeding and disease resistance mechanism research. The manuscript is well written but contains some minor conclusions that are not well supported by evidence and thus should be presented as part of the discussion, and not as part of the results. We recommend this manuscript for publication in Nature Communications after a minor revision.

Comments and questions,

1. Lines 48-50: please add a relevant reference to support this statement, such as <https://doi.org/10.1038/s41588-023-01396-w>.
2. Line 55: Please list the names of the cloned NLRs.
3. Lines 82: Did you mean "inoculated" with Bgt? Please rephrase.
4. Line 159: AelMLKL was driven by Maize ubiquitin promoter. Overexpression of some R genes may negatively affect plant productivity. Did you characterize the agronomic traits of these transgenic lines?
5. Line 200: Supplementary Fig. 7 – For the phylogenetic analysis, please add more detailed information, such as bootstrap, genetic divergence, and aligning method.
6. Lines 220-225: The authors are linking the resistance responses activated by Pm13 with PR genes. However, the presented results are not strong enough to make such conclusions since PRs are highly expressed also in the susceptible wheat line Chinese Spring (CS) after Bgt inoculation. Only at 18 hpi, the PR gene expression level in TA3575 showed higher expression than that in CS (supplementary Fig.9). Moreover, PR genes belong to the basal resistance responses, thus, a more logical explanation is that fungal effectors probably suppressed the expression of the PR genes in the susceptible line, rather than Pm13 is inducing the expression of PR genes. Moreover, this comparison should be done in the wild type and wild type + Pm13 transgenic line, while different genetic backgrounds (CS and TA3575) may cause some differences in PR gene expression levels.
7. Lines 230-233: The authors concluded in the results section: "likely eliminating the undesirable consequence due to linkage drag". The authors did not provide any proof to support this conclusion (i.e. characterization of agronomical traits). Therefore, we propose moving this statement to the discussion part or providing more solid results.
8. Line 237- Discussion: We are wondering if there are differences between Pm13 and other Pm genes, such as NLRs or TKPs, in their ROS and cell death responses. For example, in a recently published paper, Pm13 induced strong intraROS and cell death, that sometimes included 2-3 neighboring cells (<https://apsjournals.apsnet.org/doi/epdf/10.1094/PHYTO-07-22-0271-FI>, Fig. S4.). Did you find similar responses? Maybe you can discuss the ROS and cell death responses that were triggered by Pm13.
9. Lines 246-247: The authors did not provide solid evidence to show that the STK domain is functional. Thus, the classification as "functional" should be deleted here.
10. Line 249: We propose classifying these kinase fusion proteins according to the following recent paper: <https://www.nature.com/articles/s41588-023-01396-w>.
11. Lines 266-269: The fact that "all key conserved residues based on sequence alignment" of the STK kinase domain are present does not provide solid proof that the kinase is functional. To prove that the kinase is functional the authors should conduct an enzymatic bioassay and show phosphorylation

activity.

12. Lines 282-285: The author state: "It is extremely tedious to fine-map disease resistance genes like Pm13 from distant wheat wild relatives due to low exchange and recombination suppression between the alien chromatin and wheat homoeologous counterpart". This statement is valid only when attempting to clone the gene after its introgression into the hexaploid wheat background. The mentioned obstacles might be eliminated by cloning the gene directly from the diploid wild relative. Please discuss both approaches and provide examples of recently cloned kinase fusion proteins from other Agilops species.

13. Lines 366-367: Something is missing in this sentence. Please rephrase.

14. Have you done cellular localization of the Pm13 by using wheat protoplasts or a similar method?

15. Lines 319-320: The authors state: "This resistant introgression line will eliminate or decrease the undesirable linkages associated with Pm13", however, they did not provide solid results, such as characterization of agronomic traits, to support this statement. Please rephrase as a suggestion, not as a definite conclusion.

Reviewed by Tzion Fahima and Yinghui Li

Reviewer #4:

Remarks to the Author:

Li et al. describe the cloning of a new wheat resistance gene, Pm13 involved in resistance against powdery mildew. Very interestingly, this gene encodes an unusual resistance protein specific to wild wheat species from the S section. They provided markers and genetic resources to use this gene in breeding programs and improve wheat resistance against powdery mildew. This paper represents a huge amount of work to fine-map a gene from an introgression of a wild species. It emphasizes the significance of studying wild wheat species, expanding the repertoire of wheat-resistant genes, and potentially enabling the identification of a novel resistance mechanism. The findings are supported by very robust data and analyses.

I have few comments:

It is not mention anywhere if the resistance is dominant or semi dominant.

In the materials section, it is necessary to provide further details about samples, methods used for the different expression analyses (number of leaves per replicate...).

It is not clear to me if the 10 hexaploid wheat genomes have been used to search for Pm13 homologs. I would recommend checking, which will confirm the specificity of this gene.

I think that the discussion could be improved as suggested below.

Further minor comments:

L33 I would change the last sentence as I guess this is not the right message if you want to keep the gene durable.

L46/47 Please provide references.

L49/50 As written, it looks like the proteins like kinases will provide more durable resistance than NBS-LRR, which from my point of view has not been validated. Please modify and provide references.

L51 I suggest saying "could" instead of "will" as even though kinase and NLR are of different natures their resistance mechanisms could be similar and they won't be any improvement in durability.

L54 I think the references are not the right ones

L55 Please provide references and the exact number of NLR?

L120 It seems that no Methods are provided for this experiment. Please correct.

L129 Could you please provide how the annotation was performed.

L130 When you said "structurally resembling" does it means that you have look at the 3D structure, if yes please provide methods.

L130 please provide spelling for MLKL

L134 Please mention that you identified 14 susceptible families after phenotyping with isolate E09

L141 what is the Brace region, it is called domain in your figure but not colored.

L158 did you used the full length cDNA or the CDS as indicated in your Fig3. Please correct.

L160 I'm a bit confused with the sentence, do you have 14 or 20 T0?

L160 according to your Sup Fig 4 there are different levels of expression of Pm3a in your transgenic. Could you please mention if this impacts the resistance level or anything else on the plants.

L163 the Pm13a gene. Instead of locus.

L170 How can you say genetically divergent as we do not any data regarding these isolates.

L198 200 what do you suggest here, because Pm13 homologs involved in resistance to biotrophs have also a kinase domain with conserved residues, this means that the kinase is important for the resistance? Could you please explain.

L201 you cannot say functional domains as they were not functionally validated.

L224 it could be that the expression of PR genes is triggered without Pm13 as CS and TA3575 backgrounds are completely different. So please adjust rephrase.

L227 there should be a detailed material and method section on the PCR with diagnostic markers so that everyone can exactly reproduce your experiment and evaluate their materials. Could you please explain how you know that the 180 lines lack Pm13.

L241 I'm not sure to understand the link between the identification of kinase and the possibility of pyramiding, this could have been done before the identification of kinase?

L247 a most likely functional it was not experimentally validated.

L254 I like this discussion about the possible resistance mechanism of PM13 but I think that it would great to compare with the mechanism recently identified for NBS-LRR genes and the resistosome. The structure of the gene also questions me on how the gene will recognize the pathogen as it does not carry a domain like the LRR domain involved in recognition.

L271 I would suggest removing broad spectrum as the mutant lines were evaluated only against one isolate.

L290 In this paragraph and the next one you discussed the difficulties of fine-mapping genes coming from wild species and accessions without reference genomes. While I understand the significance of the use of ph line in reducing introgression for its use in breeding programs, I am uncertain whether it is the optimal and fastest method for cloning genes from wild relatives, considering the current literature and the availability of new genomic tools. I would suggest elaborating on this.

L301 This is not only the transcriptome but also the genetic mapping of contig issued from the transcriptome which is labor intensive. I suggest again here to compare with new strategies.

L315 I was unable to find reference 67 but in reference 29 the durability of Pm13 was mentioned based on a personal communication. While I acknowledge the importance of this communication, accurately assessing the long-term sustainability of this gene remains challenging based solely on this comment. I believe it is essential to exercise caution when discussing the durability of this gene and consider a more cautious approach. Especially as in the following sentence you mention that the gene was not deployed widely due to deleterious linkage drag. If it wasn't deployed largely it is not possible to really evaluate its durability.

L327 functional !

Could you please provide the generations of the recombinants.

L351 Could you please provide more information on powdery mildew isolate or provide a reference.

L356 Please provide which generation of recombinants were phenotyped?

P364 Provide the number of replicates.

L376 please provide the names of the 4 markers

L381 Please provide how markers were amplified and visualized.

L384 Could you please explain how the heterozygous recombinants were identified as your markers are dominant.

L411 Please provide the generation of the mutant used for sequencing and how cDNA were obtained.

L431 "with little homology in the wheat assemblies" What does it mean, please explain.

L439 Could you please provide the light used for this experiment.

L441 15 plants were infected but if I understand only 3 were used for checking expression. Could you please confirm?

L445 Could you please detail how the full length cDNA was cloned and the exact size/start/end of the full length cDNA

L462 Please mention how many leaves were included per biological replicates.

L473 Could you please provide further detail about the sequencing, the MasterMix used and the method of sequencing.

L488 Please provide the exact references for each genome. Could you please also define what you considered as Pm13 homologs, what should be the minimum percentage of identity, and the length of alignment....

Figure 1: Please provide the reference genome used to anchor the markers and calculate the size in Mb of the physical interval.

Figure 2 c please indicate the time point

Figure 5 please provide the name of the statistical test. There is no description of the samples used in b, at which zedock stage roots were sampled? In which conditions? The same for all other samples.

Please provide these details either here or in the Materials section. When using the delta delta Ct method, you need a control and treated samples. I suspect that you have in c, plants inoculated with Bg strain and plant treated with water but what about in b? What are your controls?

Sup Fig2 please provide the reference genome used for physical position. Please indicate the isolate. And the generation of recombinants.

Sup Fig3 Please the reference for the Ladder

Sup Fig4 Could you please let us know if the T1 plants used for Pm13 expression were selected before by PCR for being positive. It is difficult to understand if the expression is based on 3 leaves from one experiment or one leaf from 3 different experiments. Please add in the legend how the relative expression was calculated.

Sup Fig6 Could you please provide a reference for the 14 key residues of protein kinase. It would have been great to have AtMLKL in the alignment to see the non-conservation of some important residues.

Sup Fig10 Please provide a reference for your marker or at least indicate the size of the bands on the gel.

Supplementary table 1: Please provide the version of the Chinese spring genome and the reference for the Ae. Longissima genome

Supplementary table 2: I guess there is an inversion between Markers and Materials name. The title indicates 44 markers whereas in the text it is 43. Please correct. Could you please explain what are R1B and R2B. and add which isolate was used.

Supplementary table 3: same comments as ST1.

Supplementary table 4: Could you please have it the same way as ST2. Same comments as ST2.

Supplementary table 11: Please provide reference and version of the genomes. Are homologues present in syntenic position compared to Pm13?

Supplementary table 12: Could you please provide if these accessions are available from a genetic resource center.

REVIEWER COMMENTS

Reviewer #1 (Remarks to the Author):

Li et al. present their work on cloning the wheat powdery mildew resistance gene *Pm13* that is derived from *Aegilops longissima*. Several approaches were integrated to identify a mixed lineage kinase-like (MLKL) protein as the causal gene including a forward genetic screen using mutagenesis, transient virus induced gene silencing, and transgenic complementation. Comparison of *Pm13* with other known MLKL family members found structural similarity despite substantial sequence divergence. As part of this work, the authors develop a small introgression of *Pm13* and provide breeders markers for the deployment of this gene in elite cultivars.

We thank you for your comments and positive assessment of the manuscript.

Major concern 1. The identification of an MLKL encoding gene as causal for disease resistance has not been described to date. Previous work by Mahdi et al. (2020) identified the role of MLKL proteins in plant immunity and cell death processes, but their work was not based on natural genetic variation and only identified a small number of proteins described as MLKLs (typically 2 to 3 per species). A preliminary bioinformatic analysis of AtMLKL1, AtMLKL2, and AtMLKL3 found that the N-terminal domain is classified as a DUF1221, whereas *Pm13* is classified as carrying an MLKL N-terminal domain (CDD cd21037). While the authors have used AlphaFold2 to infer the potential structural relationship of *Pm13* and AtMLKL1, there lacks a complete analysis of the presence of this domain structure in other plant species. The CDD database (<https://www.ebi.ac.uk/interpro/entry/cdd/CD21037/>) suggests that over 1k proteins carry the cd21037 domain in grasses, indicating that this overall family may be larger than previous described. The current work does not sufficiently place the impact of the discovery within the context of gene family evolution (specifically, MLKL_NTD-PK). This work has several implications:

A. It appears that the original finding by Mahdi et al. underestimated the number of MLKLs in plants. Therefore, the authors should characterize this extended set of proteins that may belong to the MLKL family.

B. The existing phylogenetic tree built for the protein kinase domain is insufficient. No bootstrapping has been performed and the domains themselves have not been placed within the overall context of protein kinase domains in angiosperms. See “Diversity, classification and function of the plant protein kinase superfamily” by Melissa D. Lehti-Shiu and Shin-Han Shiu for a protocol on classifying the protein kinase domain in *Pm13*.

C. It was unclear which data sets that the authors have investigated to identify the presence of *Pm13* in the Triticeae. I suggest a Supplemental Table that documents the presence or absence in diverse data sets analyzed. I noted that *Pm13* appears to be natural varying in *Aegilops bicornis*, *Aegilops longissima*, and *Aegilops sharonensis*. I suggest the authors investigate publicly available RNAseq data sets in NCBI.

Reply: A. To explore the potential structural relationship between MLKLs identified by Mahdi et al. (2020) and *Pm13*, we identified an additional 1,435 MLKL_NTD domain-containing (CDD cd21037) proteins and 100 DUF1221 domain-containing proteins from *Poaceae* and *Arabidopsis*. Among these, 714 MLKL_NTD-Kinase proteins (Supplementary Table 11) and 97

DUF1221-Kinase proteins (Supplementary Table 12) were identified, respectively. We constructed a phylogenetic tree using these 811 proteins, together with three AtMLKL proteins (AtMLKL1, AtMLKL2 and AtMLKL3) reported by Mahdi et al. (2020) and Pm13. The dendrogram revealed that these 815 proteins were classified into two distinct clusters: namely MLKL_NTD-Kinase cluster and DUF1221-Kinase cluster. The three AtMLKLs belong to DUF1221-Kinase cluster where AtMLKL1 and AtMLKL2 in subfamily I and AtMLKL3 in subfamily II, consistent with the Mahdi's results. Pm13 was classified into the MLKL_NTD-Kinase cluster, however, it was in a subfamily distinct from any other MLKL proteins (Supplementary Fig. 8).

The amino acid sequences of 714 MLKL_NTD-Kinase proteins and 97 DUF1221-Kinase proteins were provided in Supplementary Tables 11 and 12 in revised manuscript. The phylogenetic tree based on the 815 proteins was provided in Supplementary Fig. 8 in revised manuscript.

Reference:

Mahdi, L. K. et al. Discovery of a family of mixed lineage kinase domain-like proteins in plants and their role in innate immune signaling. *Cell Host Microbe* **28**, 813-824 (2020).

B. To investigate the evolutionary origin of the Pm13 protein kinase domain, we applied a Hidden Markov Model-based classification approach for protein kinases developed by Lehti-Shiu & Shiu (2012) to analyze the kinase domain of Pm13. The result showed that Pm13 kinase domain belongs to the DLSV (DUF26, SD-1, LRR-VIII and VWA) protein kinase subfamily in the RLK (receptor-like kinase)/Pelle family. Further, we found that at least 44 genes had more than 60% amino acid sequence similarity with Pm13 kinase domain in *Poaceae*. By phylogenetic analysis of kinase domains of Pm13 and those 44 homologs together with recently reported KFP genes (Supplementary Fig. 9), we found that the kinase domain of Pm13 and 44 amino acid sequence similar genes belong to the LRR_8B subfamily (cysteine-rich receptor-like kinases), which is the most frequent kinase subfamily found in KFPs.

Reference:

Lehti-Shiu, M. D. & Shiu, S.-H. Diversity, classification and function of the plant protein kinase superfamily. *Philos Trans R Soc Lond B Biol Sci.* **367**, 2619-2639 (2012).

C. We performed a BLASTP analysis with Pm13 across the whole-genomes of common wheat *T. aestivum* (cv. Chinese Spring), *T. urartu* (cv. G1812), *Ae. tauschii* (cv. AL8/78), *T. turgidum dicoccoides* (cv. Zavitan), *T. turgidum durum* (cv. Svevo), *Hordeum vulgare* (cv. Morex), *Secale cereale* (cv. Weining), *Thinopyrum elongatum* (cv. D-3458), and five Sitopsis species of *Aegilops* (*Ae. bicornis* cv. TB01, *Ae. longissima* cv. TL05, *Ae. searsii* cv. TE01, *Ae. sharonensis* cv. TH02, and *Ae. speltoides* cv. TS01) at WheatOmics (<http://202.194.139.32/blast/blast.html>). Based on phylogenetic analysis and synteny analysis using five *Ae. longissima* genes on either side of *Pm13* across the Triticeae tribe, we found three *Pm13* orthologous genes with intact MLKL_NTD-Kinase domain (TB01.3S01G0016800.1 from *Ae. bicornis* TB01, TH02.3S01G0013600.2 from *Ae. sharonensis* TH02 and TS01.3B01G0036300.1 from *Ae. speltoides* TS01) that were present only in the S-genome of *Aegilops* species in the Triticeae tribe.

Further, we downloaded the RNA-seq raw reads of the *Aegilops* species, including three *Ae. longissima* accessions (KU-14635, KU-14624, and KU-5752), two *Ae. bicornis* accessions

(KU-14613 and KU-5784), three *Ae. sharonensis* accessions (KU-14668, KU-14663, and KU-14661), four *Ae. searsii* accessions (KU-14651, KU-6143, KU-6142, and KU-5755), seven *Ae. speltoides* accessions (KU-7848, KU-7716, KU-2236, KU-12963a, KU-14605, KU-14601, and KU-2208A) from the NCBI sequence read archive (SRA) database. Sequence comparison of the *Pm13* sequence and RNA-seq raw reads of the *Aegilops* species, we found only *Ae. longissima* accession KU-14635 had identical cDNA sequence with *Pm13*. No reads or few reads from RNA-seq databases of other *Aegilops* species demonstrated an alignment on *Pm13* cDNA sequence, suggesting likely lacking the *Pm13* allele or weak expression of *Pm13* allele in these reference accessions. Since we do not have these *Aegilops* species to test these possibilities, we have not added this information in the manuscript.

Major concern 2. The authors specifically state that the protein kinase domain of Pm13 is functional but they provide no molecular evidence for this claim (such as lines 200-202). The text either needs to sufficiently temper this statement (such as “Pm13 is predicted to be a functional protein kinase based on the presence of key residues for kinase activity”) or kinase assays need to be performed. Regardless, the authors need to prepare a supplementary figure where they show the position of the conserved residues for kinase activity.

Reply: According to Hanks et al. (2008), protein kinases rely on the Lys of the Val-Ala-Ile-Lys (VAIK) motif to interact with the α and β phosphates of ATP to position ATP during phosphotransfer, the Asp of the His-Arg-Asp (HRD) motif within the catalytic loop (subdomain VIb) to act as a catalytic residue, and the Asp within the Asp-Phe-Gly (DFG) motif in the activation loop (subdomain VII) to bind Mg^{2+} to coordinate the β and γ phosphates of ATP. The kinase domain of Pm13 presents all three conserved catalytic residues that are crucial for phosphoryl transfer activity, thus we inferred that Pm13 kinase domain might also be a functional protein kinase based on the presence of key residues for kinase activity. The position of the conserved residues for kinase activity was presented in Supplementary Fig. 7 in revised manuscript. However, we agree with reviewer’s suggestions that whether Pm13 kinase domain is functional or not should be proved by kinase function assays. Accordingly, we have updated the statement as suggested to “The C-terminal kinase domain of Pm13, in contrast to pseudokinase domains of the mammalian MLKL and *A. thaliana* AtMLKLs, presented all key conserved residues for kinase activity based on sequence alignment, indicating that Pm13 is likely a functional protein kinase.” in lines 218-220 in revised manuscript.

Reference:

Hanks, S. K., Quinn, A. M., and Hunter, T. The protein kinase family: conserved features and deduced phylogeny of the catalytic domains. *Science* **241**, 42-52 (1988).

Major concern 3. A statement about broad-spectrum resistance has to be tempered with the diversity of isolates used for assessment. In this work, the isolates are predominantly only from China (although I believe reference 29 tested a single Italian isolate), therefore I suggest the authors temper their statements and let the passage of time show the utility of Pm13.

Reply: According to Wu et al. (2018), *Pm13* was resistant to 39 *Bgt* isolates collected from different wheat fields located in the provinces of Hebei, Shandong, Henan, Beijing and Jiangsu, China, indicating that *Pm13* may be a putative broad-spectrum resistance gene. In the revised manuscript, we deleted “broad-spectrum”.

Reference:

Wu, P. P. et al. Development of molecular markers linked to powdery mildew resistance gene *Pm4b* by combining SNP discovery from transcriptome sequencing data with bulked segregant analysis (BSR-Seq) in wheat. *Front. Plant Sci.* **9**, 95 (2018).

Minor concerns

1. Throughout, the references appear to be incorrect in several places.

Reply: We have checked and corrected the references in revised manuscript.

2. Throughout, CS was not defined as Chinese Spring.

Reply: We have rectified this error and defined the abbreviation at first instance.

3. Abstract, “due to linkage drag” to “due to linkage drag with deleterious traits”.

Reply: This has been changed as you suggested in line 32 in revised manuscript.

4. Abstract, “The new lines...” to “Our sequential reduction of the genomic segment encompassing *Pm13* will facilitate wide-scale deployment in wheat”.

Reply: The statement has been modified as you suggested in lines 32-36 in revised manuscript.

5. Introduction, lines 48-49, the language of the sentence is unclear. Perhaps consider “Recent work has shown that the Triticeae contain an extended array of resistance genes that include diverse kinase domains.” I would avoid associating the presence of a kinase domain with durability, as this has not been shown in any conclusive systematic manner.

Reply: This has been changed as you suggested in lines 50-52 in revised manuscript.

6. Introduction, line 73, “at chromosome” to “located on chromosome” also “at 6S” to “located on 6S”.

Reply: This has been changed as you suggested in line 76 in revised manuscript.

7. Introduction, line 80, “by a strategy” to “using a strategy”.

Reply: We have updated the statement as you suggested in line 83 in revised manuscript.

8. Results, line 91, “In absence” to “In the absence”.

Reply: This has been changed as you suggested in lines 94-96 in revised manuscript.

9. Results, line 92, “reference genome we performed transcriptome sequences to” to “reference genome, we performed transcriptome sequencing to”.

Reply: This has been changed as you suggested in line 95 in revised manuscript.

10. Results, line 124, “Thus, gene” to “Thus, the gene”.

Reply: This has been changed as you suggested in line 128 in revised manuscript.

11. Results, lines 133 to 139. The authors identify 14 loss-of-function mutants but only provided

discussion on the 11 loss-of-function mutants in *Pm13*. The authors do not comment on complementation tests (*Pm13* and non-*Pm13* mutants), nor the potential role of these mutants in elucidating *Pm13* function.

Reply: By comparing both cDNA and gDNA sequences of gene *Pm13* with the other three loss-of-function EMS mutants, we didn't find any sequence variation on the sequence of *Pm13* gene. Therefore, these three mutations may have sequence variation on other genes involving in the *AelMLKL* regulation pathway. We are currently making crosses between *Pm13* wild type and mutants and these non-*Pm13* mutants for genetic analysis and complementation tests. We have added the statement "No sequence variation was detected for the other three EMS-induced susceptible mutants in the entire *AelMLKL* gene, suggesting possible mutations in other unknown genes involved in the *AelMLKL* regulation pathway." in lines 155-157 in revised manuscript.

12. Discussion, line 315. Change "However, the gene was deployed only in a few wheat..." to "However, the gene had limited deployment in wheat...".

Reply: This has been changed as you suggested in lines 381-382 in revised manuscript.

13. Discussion, lines 319-320. I suggest tempering this sentence, as the authors do not show if this line does not suffer deleterious effects in the field. I agree that the line is promising.

Reply: We have tempered this sentence as "This resistant introgression line could minimize the undesirable linkages associated with *Pm13*." in lines 385-386 in revised manuscript.

14. Discussion, line 330. Change "tiny" to the actual size.

Reply: This has been changed "the resistant introgression line harboring *Pm13* with 2.82 Mb of 3S¹#2 genomic length" in line 395 in revised manuscript.

Reviewer #2 (Remarks to the Author):

In this manuscript, Li and colleagues cloned the powdery mildew resistance gene *Pm13* that was introgressed into bread wheat from the wild wheat relative *Aegilops longissima*. The authors use genetic mapping, mutagenesis, gene silencing and transgenic complementation and show that *Pm13* encodes an unusual kinase fusion protein. The manuscript adds to the growing number of kinase fusion proteins involved in disease resistance in Triticeae. The study is well executed and of interest to the broad readership of Nature Communications, contributing novel insights into one of the hottest topics in the field of host-pathogen interactions in cereals (the role of unusual kinase fusion proteins in plant-pathogen interactions). I do have some comments that need to be addressed before publication:

We thank you for your comments and positive assessment of the manuscript.

Major comments:

- For the transformation, the authors used an overexpressing construct (Ubi promoter) and the *Pm13* coding sequence. There are several examples of disease resistance genes that have possibly been wrongly identified in wheat because of candidate gene overexpression. A critical experiment is to demonstrate that the overexpressing transgenic lines maintain race-specificity. According to literature, virulence to *Pm13* has been reported (for example Zhang et al. 2022 PeerJ 10: e14118). Ideally, the authors can test a virulent *Bgt* isolate on some of their transgenic lines to prove that the resistance seen in the transgenic lines is not caused by a general activation of immunity by the overexpressing the *Pm13* CDS.

Reply: Zhang et al. (2022) reported that *Bgt* isolate *VI3* was virulent to *Pm13*. We contacted the authors and was informed that *Bgt* isolate *VI3* was not maintained and all the authors of the publication no longer engaged in evaluation of wheat powdery mildew resistance. We also contacted other *Bgt* research labs from Beijing, Sichuan, Shandong and Henan provinces, and couldn't find any *Bgt* isolates virulent to *Pm13*. To exclude the possible incorrect identification of the candidate resistance gene caused by R gene overexpression, we have re-examined the *Pm13* expression levels in all the positively transgenic T₁ lines. We noticed that although the transgenic line *AelMLKL-OE3* showed the similar *Pm13* expression level as *Pm13* donor parent TA3575, the plants of this transgenic line still conferred high resistance to *Bgt* isolate E09 as TA3575. The result indicated that the resistance conducted by the *Pm13* transgenic lines were not resulted from a general activation of immunity by overexpressing the *Pm13* CDS. In addition, the 11 of the 14 EMS-induced mutants losing powdery mildew resistance mutated in *AelMLKL* sequences also be a strong proof of the candidate gene *AelMLKL* is the *Pm13*.

- The authors state that the HeLo-kinase fusion is unique to Sitopsis species of the genus *Aegilops* and absent in bread wheat and other wild wheat relatives. This would indicate a rather recent fusion event. The manuscript could be greatly improved by adding additional information regarding the possible evolution of *Pm13*. For example, the authors show in Supplementary Fig. 7 that the *Pm13* kinase domain is similar to kinase domains of other kinase fusion proteins involved in disease resistance, including *Pm4b* and *WTK2*. But what about the N-terminal HeLo domain? What are the closest homologs in bread wheat when only using the HeLo domain? Are HeLo

domains usually fused to other domains in bread wheat and other Triticeae species? This could help to make predictions about the possible function of Pm13. For example, is it possible that the HeLo domain in Pm13 is an integrated decoy?

Reply: The Pm13 contains a MLKL-NTD domain in its N-terminus and a C-terminus STK domain based on the conserved domain database from the National Center for Biotechnology Information (NCBI). The 3D model revealed that the N-terminal MLKL-NTD domain of Pm13 consisted of a four-helix bundle (4HB) structure, also known as the HeLo domain, thus the N-terminal MLKL-NTD domain of Pm13 was designated as HeLo domain in original manuscript.

Mahdi et al. (2020) identified a limited number of MLKL proteins and the N-terminal domain of these MLKLs was classified as a DUF1221 domain. The DUF1221 domain possessed a four-helix bundle (4HB), like HeLo domain and MLKL-NTD domain.

To clear the confusion caused by different classification, we renamed the HeLo domain of Pm13 described in previous manuscript version to MLKL_NTD domain, and classified Pm13 as a MLKL_NTD-Kinase protein in revised manuscript.

We analyzed the relationships of MLKL_NTD domain (also named HeLo domain) of Pm13 using only the HeLo domain. We firstly identified 220 additional MLKL_NTD domain-containing proteins across the Triticeae tribe at WheatOmics (<http://202.194.139.32/blast/blast.html>) (Supplementary Table 14), which included 158 (71.82%) MLKL_NTD-Kinase proteins, 50 (22.73%) MLKL_NTD proteins, 8 (3.64%) MLKL_NTD-MLKL_NTD proteins, 3 (1.36%) MLKL_NTD-WD40 proteins, and one (0.45%) MLKL_NTD-Kinase-Kinase protein (Supplementary Table 15).

Subsequently, we constructed a phylogenetic tree based on these 220 MLKL_NTD domain-containing proteins (Supplementary Fig. 10), and a second phylogenetic tree based on 228 MLKL_NTD domains (Supplementary Table 16) from 220 MLKL_NTD domain-containing proteins (Supplementary Fig. 11). The two dendrograms displayed that the closest homologs of Pm13 were three proteins that include TB01.3S01G0016800.1 from *Aegilops bicornis* TB01, TH02.3S01G0013600.2 from *Ae. sharonensis* TH02, TS01.3B01G0036300.1 from *Ae. speltoides* TS01. These proteins were all MLKL_NTD-Kinase proteins and shared 99.37%, 97.27% and 89.94% sequence identity with Pm13 (Supplementary Fig. 12), respectively. Phylogenetic analysis indicated that Pm13-like proteins likely only exist in S-genome of *Aegilops*. However, no closest homologs were discovered in bread wheat and other Triticeae species except S-genome species. These results suggest that Pm13 gene may originate after the formation of the S-genome species. We have added these results in lines 247-258 in revised manuscript.

The amino acid sequence of 220 MLKL_NTD domain-containing proteins were provided in Supplementary Table 14 in revised manuscript.

The classification of 220 MLKL_NTD domain-containing proteins were provided in Supplementary Table 15 in revised manuscript.

The amino acid sequences of 228 MLKL_NTD domains from 220 MLKL_NTD domain containing proteins were provided in Supplementary Table 16 in revised manuscript.

The phylogenetic tree based on 220 MLKL_NTD domain containing proteins were provided in Supplementary Fig. 10 in revised manuscript.

The phylogenetic tree based on 228 MLKL_NTD domains from 220 MLKL_NTD domain containing proteins were provided in Supplementary Fig. 11 in revised manuscript.

The protein sequence alignment of Pm13 and Pm13 orthologs were provided in Supplementary Fig. 12 in revised manuscript.

Minor comments:

- Abstract: ‘Thus, Pm13 represents a special class of plant resistance genes revealing a novel aspect of host-pathogen interactions.’ Pm13 definitely has a novel and unusual domain architecture, but I would not call it a ‘special class’ that reveals a ‘novel aspect of host-pathogen interactions’. A good number of Triticeae disease resistance genes encode kinase fusion proteins and they probably all follow a similar molecular mechanism. This has recently been discussed (<https://www.researchsquare.com/article/rs-1820134/v1>), resulting in the integrated decoy model. I think that the Pm13 gene might fall into this category of kinase fusion proteins and should be discussed with respect to previous findings (see also my comment above regarding a deeper evolutionary analysis of the two domains).

Reply: To avoid confusion, we have deleted “Thus, Pm13 represents a special class of plant resistance genes revealing a novel aspect of host-pathogen interactions.” in Abstract in revised manuscript.

- Abstract: The authors state that Pm13 has not been widely used in breeding. I would thus refrain from using the term ‘broad-spectrum’. In my view, this term should only be used if a disease resistance gene maintains broad-spectrum specificity when exposed to the pathogen in agriculture.

Reply: We have revised the text as suggested to avoid statement of “broad-spectrum” resistance.

- Line 43: ‘can lead to an estimated yield loss of up to 40%’. This is true, but a more realistic estimation of the average yield losses caused by powdery mildew is around 1% (Supplementary Table 3 of Savary et al. 2019 Nature Ecology & Evolution 3: 430-439).

Reply: We have revised as you suggested in line 45 in revised manuscript.

- Lines 48-52: I don’t think that there is any evidence that kinase fusion proteins are more durable compared to NLR-mediated immunity. Also, it is not clear if kinase fusion proteins function through a ‘different molecular mode’ as suggested here. This statement is misleading because it suggests that kinase fusion proteins provide more durable resistance and function through a novel molecular mechanisms, for which there is no evidence yet.

Reply: We have updated the statement as “Recent research has shown that the Triticeae contain an extended array of resistance genes that include diverse kinase domains, which are new players in plant immunity. Therefore, cloning novel types of powdery mildew resistance (*Pm*) genes could facilitate their fast-tracking in breeding programs and incorporation into polygene stacks for maximizing the durability of powdery mildew resistance.” in lines 50-54 in revised manuscript.

- Line 70-71: What does the nomenclature 3S¹#1S, 3S¹#2S, etc. refer to? Is this the same chromosome from different *Ae. longissima* accessions?

Reply: The nomenclature 3S¹#1S, 3S¹#2S and 3S¹#3S refer to the same chromosome 3S¹ from three different *Ae. longissima* accessions. The numbers 1, 2, 3 are used to represent three different accessions, and “S” following the numbers 1, 2, 3 represents the short arm of the chromosome.

The donor of chromosome 3S¹1S in Chinese Spring (CS)-*Aegilops longissima* T3BL.3BS-3S¹1S translocation line R1B is derived from *Ae. longissima* accession TL01 (Biagetti et al. 1999). The donor of chromosome 3S¹2S in CS-*Ae. longissima* 3S¹2(3B) disomic substitution line TA3575 is derived from *Ae. longissima* accession TL20 (Feldman 1975). The donor of chromosome 3S¹3S in CS-*Ae. longissima* 3S¹3 isochromosome addition line TA7545 is derived from *Ae. longissima* accession TA1910 (TAM4) (Friebe et al. 1993).

Reference:

Biagetti, M., Vitellozzi, F. & Ceoloni C. Physical mapping of wheat-*Aegilops longissima* breakpoints in mildew-resistant recombinant lines using FSIH with highly repeated and low-copy DNA probes. *Genome* **42**, 1013-1019 (1999).

Feldman, M. Allen addition lines of common wheat containing *Triticum longissimum* chromosomes. *Proc. 12th Int. Bot. Congr.* **22**, 237-259 (1975).

Friebe, B., Tuleen, N., Jiang, J. M. & Gill, B. S. Standard karyotype of *Triticum longissima* and its cytogenetic relationship with *T. aestivum*. *Genome* **36**, 731-742 (1993).

- Line 91-92: Obviously, an *Ae. longissima* reference genome was produced during this project, but was not available at the beginning of this project. This can be slightly confusing. The sentence here could be re-phrased to be more clear. For example ‘Because no *Ae. longissima* reference genome was available at the beginning of this project, we used transcriptome sequencing to design a set of 43 3S¹2-specific markers.’

Reply: We have revised as you suggested in lines 94-96 in revised manuscript.

- Lines 168-169: Please provide the number of populations and the number of F₂ plants used for the allelism tests. The numbers are provided in Supplementary Table 9, but it would be beneficial for readers to also have them in the main text.

Reply: We have provided provided the requested numbers for the allelism tests in lines 182-187 in revised manuscript.

- Line 175: I suggest writing ‘thus providing strong evidence that *Pm13*, *Pm13a*, and *Pm13b* are the same gene (*Pm13* hereafter)’.

Reply: We have revised as you suggested in line 193 in revised manuscript.

- Line 186: ‘confirming that both domains and connection brace are required...’ should be replaced with ‘indicating that both domains and connection brace might be required’. It is also possible that the amino acid substitutions affect protein stability and protein abundance as a whole.

Reply: We have revised as you suggested in line 211 in revised manuscript.

- Line 239: ‘A wide majority of the R genes cloned till date encode NLRs’. I would argue that this statement is probably no longer true. The number of cloned kinase fusion proteins in Triticeae increased to 17 with the addition of *Pm13* and *Pm57*, which is still less compared to NLR-encoding genes, but NLRs are no longer a ‘wide majority’ in Triticeae.

Reply: We have changed the wording as “Around 81 Triticeae resistance genes have been cloned till date, the common classes encoded NLRs” in lines 290-291 in revised manuscript.

- Lines 241-243: Here, the authors imply again that kinase fusion gene stacks might be more durable compared to NLR gene stacks. There is no experimental evidence for this and this statement should be rephrased.

Reply: We have rephrased the statement as “Recently resistance genes encoding KFPs have expanded the repertoire of non-NLR genes in the Triticeae tribe for resistance breeding.” in lines 291-293 in revised manuscript.

- Lines 248-252: Although Pm13 has a different domain architecture compared to other kinase fusion proteins, it is highly unlikely that Pm13 functions through a ‘special disease resistance mechanism’. Two preprints recently suggested a common model for kinase fusion proteins (<https://www.researchsquare.com/article/rs-1820134/v1> and <https://www.researchsquare.com/article/rs-1807889/v1>) named the integrated decoy model. I think it would make more sense to discuss Pm13 in the context of this model, rather than suggesting a new molecular mechanism (see also my major comment regarding the evolutionary history of Pm13).

Reply: *Pm13* encodes a novel kinase fusion protein (KFP) that contains a kinase domain in its C terminus fused to a MLKL_NTD domain in N-terminal, distinct from other cloned KFPs. Based on MLKL_NTD domain, Pm13 can also be classified as MLKLs. According to Yu et al. (2023), Wang et al. (2023) and Fahima & Coaker (2023), KFPs have one apparent functional kinase that is fused to a second non-functional kinase domain or an entirely different domain. The integrated domains in KFPs were hypothesized to perceive pathogen effectors, while the kinase catalyze the phosphorylation of either the effector or the NLR guard to initiate downstream defense responses and immunity. In contrast, the MLKL_NTD domain in MLKLs proteins were reported to relate to cell death, and the kinase domain of MLKLs were nonfunctional pseudokinase domain. Therefore, whether the integrated MLKL_NTD domain presenting in *Pm13* may act as decoys of virulence effector proteins delivered into plant cells, and whether the kinase domain of Pm13 may activate a downstream defense responses and immunity like other KFPs needs to be proven. We are conducting researches to give insight into molecular mechanisms of *Pm13* by experimental proofs. We have discussed the possible disease-resistant mechanism of Pm13 in the context of a common model for kinase fusion proteins suggested by Yu et al. (2023), Wang et al. (2023) and Fahima & Coaker (2023) in lines 296-309 and lines 324-328 in revised manuscript.

Reference:

- Yu, G. T. et al. The wheat stem rust resistance gene Sr43 encodes an unusual protein kinase. *Nat Genet.* **55**, 921-926 (2023).
- Wang, Y. J. et al. An unusual tandem kinase fusion protein confers leaf rust resistance in wheat. *Nat Genet.* **55**, 914-920 (2023).
- Fahima, T., & Coaker, G. Pathogen perception and deception in plant immunity by kinase fusion proteins. *Nat Genet.* **55**, 908-909 (2023).

Reviewer #3 (Remarks to the Author):

Li et al. reported the map-based cloning of broad-spectrum powdery mildew resistance gene *Pm13* from a wild wheat relative *Aegilops longissima*. The cloning method and proofs are solid and convincing. The candidate gene was validated by loss-of-function mutants, gene silencing, transgenic assay, and allelic association analyses. Interestingly, *Pm13* encodes a novel mixed lineage kinase domain-like (MLKL) protein which contains an N-terminal HeLo domain and a C-terminal authentic serine/threonine kinase (STK) domain, a unique domain architecture that has not been reported before for plant disease-resistance genes. Cloning of *Pm13* will benefit wheat breeding and disease resistance mechanism research. The manuscript is well written but contains some minor conclusions that are not well supported by evidence and thus should be presented as part of the discussion, and not as part of the results. We recommend this manuscript for publication in Nature Communications after a minor revision.

We thank you for your comments and positive assessment of the manuscript.

Comments and questions,

1. Lines 48-50: please add a relevant reference to support this statement, such as <https://doi.org/10.1038/s41588-023-01396-w>.

Reply: This has been changed accordingly in revised manuscript.

2. Line 55: Please list the names of the cloned NLRs.

Reply: We have added the requested names in lines 57-59 in revised manuscript.

3. Lines 82: Did you mean "inoculated" with *Bgt*? Please rephrase.

Reply: RNA-seq of *Pm13a* donor parent TA3575 was performed inoculated with *Bgt* isolate E09.

4. Line 159: *AelMLKL* was driven by Maize ubiquitin promoter. Overexpression of some R genes may negatively affect plant productivity. Did you characterize the agronomic traits of these transgenic lines?

Reply: Due to the COVID-9 epidemic last year, the campus adopted closed management. The *AelMLKL* transgenic lines in the experimental fields were planted very late and could not be properly managed. Even though, we found no significant effects of *AelMLKL* overexpression on plant growing and development, and preliminarily investigated agronomic traits such as plant height, spike length, spikelets per spike, grain size. To ensure the reliable evaluation of the effects of *Pm13* overexpression, we are planning to propagate the transgenic lines and investigate the *Pm13* effects to the agronomic traits of the transgenic lines under different wheat production conditions in multi-plot demonstration for several years. We excluded the evaluation of the agronomic traits of these transgenic lines in the current version of the manuscript.

5. Line 200: Supplementary Fig. 7 – For the phylogenetic analysis, please add more detailed information, such as bootstrap, genetic divergence, and aligning method.

Reply: We conducted a series of additional experiments, Supplementary Fig. 7 has changed to Supplementary Fig. 9. We have supplemented bootstrap value in Supplementary Fig. 9 and Methods “Phylogenetic analysis of *Pm13*” section in revised manuscript.

6. Lines 220-225: The authors are linking the resistance responses activated by *Pm13* with *PR* genes. However, the presented results are not strong enough to make such conclusions since *PRs* are highly expressed also in the susceptible wheat line Chinese Spring (CS) after *Bgt* inoculation. Only at 18 hpi, the *PR* gene expression level in TA3575 showed higher expression than that in CS (supplementary Fig.9). Moreover, *PR* genes belong to the basal resistance responses, thus, a more logical explanation is that fungal effectors probably suppressed the expression of the *PR* genes in the susceptible line, rather than *Pm13* is inducing the expression of *PR* genes. Moreover, this comparison should be done in the wild type and wild type + *Pm13* transgenic line, while different genetic backgrounds (CS and TA3575) may cause some differences in *PR* gene expression levels.

Reply: To reduce differences in genetic backgrounds (CS and TA3575) influence *PR* gene expression, we re-evaluated the transcript levels of six *PR* genes (*PR1*, *PR2*, *PR3*, *PR4*, *PR5* and *PR9*) in *Pm13* transgenic lines AelMLKL-0E11 and AelMLKL-0E14, and susceptible control Fielder at different time points after inoculation with *Bgt* isolate E09. As presented in Supplementary Fig. 14 in revised manuscript, both *Pm13* transgenic lines and susceptible control Fielder showed low expression levels of *PR* genes before infection. The transcript levels of *PR* genes increased after inoculation in both *Pm13* transgenic lines and susceptible control Fielder, however, the *PR* gene expression levels were significantly higher in *Pm13* transgenic lines than in susceptible control Fielder (Supplementary Fig. 14). The expression patterns indicated that *Pm13* transgenic lines induced the defense response more rapidly and stronger than susceptible control Fielder did. We have complemented the transcript levels of six *PR* genes (*PR1*, *PR2*, *PR3*, *PR4*, *PR5* and *PR9*) in *Pm13* transgenic lines AelMLKL-0E11 and AelMLKL-0E14, and susceptible control Fielder in lines 269-277 and Supplementary Fig. 14 in revised manuscript.

7. Lines 230-233: The authors concluded in the results section: “likely eliminating the undesirable consequence due to linkage drag”. The authors did not provide any proof to support this conclusion (i.e. characterization of agronomical traits). Therefore, we propose moving this statement to the discussion part or providing more solid results.

Reply: Following your suggestions, we are going to evaluate the growth and development and agronomic traits of CS-*Ae. longissima* 3S^l#2 recombinant W12-3 that contains small segments harboring *Pm13* together with CS, CS-*Ae. longissima* 3S^l#2(3B) disomic substitution line TA3575 and other *Pm13* recombinants under different wheat production conditions to investigate linkage drags. We have excluded the evaluation of the agronomic traits of W12-3 in the current version of the manuscript. Accordingly, we have changed the sentence as “we developed a T3S^l#2S-3BS.3BL recombinant W12-3 which has a small 3S^l#2 segment harboring *Pm13* in wheat background ranging from 0 to 2.82 Mb (3SI-31B), 0.35% of 3S^l length in *Ae. longissima* TL05 reference genome, which will likely minimize linkage drags.” in lines 282-285 in revised manuscript.

8. Line 237- Discussion: We are wondering if there are differences between *Pm13* and other *Pm* genes, such as NLRs or TKPs, in their ROS and cell death responses. For example, in a recently published paper, *Pm13* induced strong intraROS and cell death, that sometimes included 2-3 neighboring cells (<https://apsjournals.apsnet.org/doi/epdf/10.1094/PHYTO-07-22-0271-FI>, Fig.

S4.). Did you find similar responses? Maybe you can discuss the ROS and cell death responses that were triggered by *Pm13*.

Reply: As you suggested, we have identified the ROS and cell death responses of *Pm13* following to the methods described by Li et al. (2023), and observed at least 100 infected cells per leaf of *Pm13* donor line TA3575 and CS for each of three biological replicates. The results showed that the proportion of ROS-mediated cell death responses of *Pm13* line TA3575 were significantly higher than the susceptible line CS. The quantitative resistance responses of *Pm13* line TA3575 were identical with Li et al. (2023) and supplemented in lines 263-265, and Fig. 5b-c in revised manuscript.

Reference:

Li, Y. H. et al. Intracellular reactive oxygen species-aided localized cell death contributing to immune responses against wheat powdery mildew pathogen. *Phytopathology* **113**, 884-892 (2023).

9. Lines 246-247: The authors did not provide solid evidence to show that the STK domain is functional. Thus, the classification as “functional” should be deleted here.

Reply: We have deleted the term “functional” as you suggested in line 296 in revised manuscript.

10. Line 249: We propose classifying these kinase fusion proteins according to the following recent paper: <https://www.nature.com/articles/s41588-023-01396-w>.

Reply: According to Fahima & Coaker (2023), *Pm13* is classified to be a kinase fusion protein (KFP) that contains a kinase domain fused to a second MLKL_NTD kinase domain. Phylogenetic analysis of kinase domains of *Pm13* with recently cloned KFPs revealed that the kinase domain of *Pm13* belongs to the LRR_8B subfamily (Supplementary Fig. 9), the most frequent kinase subfamily found in cloned KFPs.

Reference:

Fahima, T., & Coaker, G. Pathogen perception and deception in plant immunity by kinase fusion proteins. *Nat Genet.* **55**, 908-909 (2023).

11. Lines 266-269: The fact that “all key conserved residues based on sequence alignment” of the STK kinase domain are present does not provide solid proof that the kinase is functional. To prove that the kinase is functional the authors should conduct an enzymatic bioassay and show phosphorylation activity.

Reply: We completely agree with the fact that the presence of all key conserved residues in the STK kinase domain of *Pm13* does not indicate the kinase is functional, and it needs experimental proof to support the conclusion. Hence, as suggested by another reviewer, we have updated this statement to “Unlike C-terminal pseudokinase domain of animal MLKLs and plant *AtMLKLs*, the C-terminal kinase domain of *Pm13* comprises conserved residues crucial for kinase activity, as evidenced by sequence alignment. Resistance loss induced by ten mutant sites from nine mutants located in the kinase domain of *Pm13* implies that it functions as a protein kinase.” in lines 303-307 in revised manuscript.

12. Lines 282-285: The author state: “It is extremely tedious to fine-map disease resistance genes like *Pm13* from distant wheat wild relatives due to low exchange and recombination suppression

between the alien chromatin and wheat homoeologous counterpart”. This statement is valid only when attempting to clone the gene after its introgression into the hexaploid wheat background. The mentioned obstacles might be eliminated by cloning the gene directly from the diploid wild relative. Please discuss both approaches and provide examples of recently cloned kinase fusion proteins from other *Aegilops* species.

Reply: We agree with you that the recombination suppression between the alien chromatin and wheat homoeologous counterpart might be eliminated by cloning the gene directly from the diploid wild relative. However, many valuable sources of disease resistance genes have already been introgressed into common wheat from its wild relatives and effective for multiple races. Cloning such important genes in wheat background is vital but map-based cloning of such genes is very difficult due to low recombination between alien chromosomes and wheat homoeologous counterparts that is suppressed by homoeologous pairing (*Ph*) loci present in common wheat. Thus, we think it is relatively tedious to isolate disease resistance genes like *Pm13* that have been introgressed into common wheat from distant wheat wild relatives. We are sorry not expressed it clearly .

However, map-based cloning of those alien genes has been accelerated by deployment of *ph1b* gene that induced homoeologous pairing and recombination, the availability of more and more genome reference sequences, the emergence of sequencing technologies, and new gene cloning approaches such as MutSeqs. Besides, wheat-wild specie recombinants with small alien segments harboring the targeted genes can be produced accompanying the cloning completion, providing resistance germplasms likely more suitable for wheat breeding program. Therefore, we think that map-based cloning of a resistance gene from wild species introgressed into common wheat is still a choice in spite of time and energy-consuming compared to other approaches.

As suggested, we have revised the statement to “As a comparison, it is relatively tedious to fine-map disease resistance genes like *Pm13* that has been introgressed into common wheat from distant wheat wild relatives like *Ae. longissima* due to low exchange and recombination suppression between the alien chromatin and wheat homoeologous counterpart that is strictly controlled by a complex *Ph* system in hexaploid wheat genetic backgrounds.”. Further we have discussed the approaches recently applied to clone resistant genes from wheat and its wild relatives in lines 335-346 in revised manuscript.

13. Lines 366-367: Something is missing in this sentence. Please rephrase.

Reply: Thanks pointing it out. We have supplemented detailed information for detecting the ROS accumulation and plant cell death in Methods “Detection of ROS accumulation and plant cell death” section in lines 428-441 in revised manuscript.

14. Have you done cellular localization of the Pm13 by using wheat protoplasts or a similar method?

Reply: In the revised manuscript, subcellular localization of Pm13 through transient transformation in wheat leaf protoplasts has been presented in Supplementary Fig. 5, the Results “Subcellular localization and structural analysis of Pm13 protein” and Methods “Subcellular localization analysis” sections. The transient expression of Pm13 and GFP fusion protein in wheat leaf protoplasts showed that Pm13 was localized in nucleus and cytoplasm.

15. Lines 319-320: The authors state: “This resistant introgression line will eliminate or decrease the undesirable linkages associated with *Pm13*”, however, they did not provide solid results, such as characterization of agronomic traits, to support this statement. Please rephrase as a suggestion, not as a definite conclusion.

Reply: Thanks for the suggestions. We will propagate seeds of the 3S^l#2 small segment recombinant W12-3 and conduct multiple-year experiments under different production conditions to investigate if the recombinant can decrease the undesirable linkages associated with *Pm13*. Accordingly, we have changed the wording as “This resistant introgression line could minimize the undesirable linkages associated with *Pm13*.” in lines 385-386 in revised manuscript.

Reviewed by Tzion Fahima and Yinghui Li

Reviewer #4 (Remarks to the Author):

Li et al. describe the cloning of a new wheat resistance gene, *Pm13* involved in resistance against powdery mildew. Very interestingly, this gene encodes an unusual resistance protein specific to wild wheat species from the S section. They provided markers and genetic resources to use this gene in breeding programs and improve wheat resistance against powdery mildew. This paper represents a huge amount of work to fine-map a gene from an introgression of a wild species. It emphasizes the significance of studying wild wheat species, expanding the repertoire of wheat-resistant genes, and potentially enabling the identification of a novel resistance mechanism. The findings are supported by very robust data and analyses.

We thank you for your comments and positive assessment of the manuscript.

I have few comments:

It is not mention anywhere if the resistance is dominant or semi dominant.

Reply: The resistance of *Pm13* was dominant in common wheat background.

In the materials section, it is necessary to provide further details about samples, methods used for the different expression analyses (number of leaves per replicate...).

Reply: As suggested, we have elaborated the Methods section in revised manuscript to provide more information related to the performed research.

It is not clear to me if the 10 hexaploid wheat genomes have been used to search for *Pm13* homologs. I would recommend checking, which will confirm the specificity of this gene.

I think that the discussion could be improved as suggested below.

Reply: We performed a BLASTP analysis across the 10 hexaploid wheat genomes (cv. Chinese Spring, cv. Fielder, cv. ArinaLrFor, cv. Julius, cv. LongReach_Lancer, cv. CDC_Landmark, cv. Mace, cv. Norin61, cv. CDC_Stanley, cv. SY_Mattis, cv. Renan, cv. Karioga, cv. Kenong9024, cv. Cadenza, cv. Claire, cv. Paragon, cv. Robigus and cv. Weebill) in WheatOmics (<http://wheatomics.sdau.edu.cn/>) and did not find any *Pm13* homologs.

Further minor comments:

L33 I would change the last sentence as I guess this is not the right message if you want to keep the gene durable.

Reply: We have changed the sentence as “Isolating of *Pm13* and development of wheat-*Ae. longissima* recombinants with significantly reduced genomic segments encompassing *Pm13* may provide valuable insights into the molecular mechanisms underlying *Pm13*-mediated powdery mildew resistance and enable widespread deployment of this gene into wheat cultivars” in lines 32-36 in revised manuscript.

L46/47 Please provide references.

Reply: We have provided references as requested in revised manuscript.

Reference:

Keller, B. & Sanchez-Martin, J. NLR immune receptors and diverse types of non-NLR proteins control race-specific resistance in Triticeae. *Curr Opin Plant Biol.* **62**, 102053 (2021).

L49/50 As written, it looks like the proteins like kinases will provide more durable resistance than NBS-LRR, which from my point of view has not been validated. Please modify and provide references.

Reply: Apologies for the imprecise expression. We have changed the sentence to read “Recent research has shown that the Triticeae contained an extended array of resistance genes that include diverse kinase domains, which are new players in plant immunity.” in lines 50-52 in revised manuscript.

L51 I suggest saying “could” instead of “will” as even though kinase and NLR are of different natures their resistance mechanisms could be similar and they won’t be any improvement in durability.

Reply: We have updated the statement using “could” instead of “will” as suggested in line 53 in revised manuscript.

L54 I think the references are not the right ones.

Reply: We have provided the right references as suggested in line 57 in revised manuscript.

References:

McIntosh, R. A., Dubcovsky, J., Rogers, W. J., Xia, X. C., & Raupp, W. J. Catalogue of gene symbols for wheat: 2020 supplement. *Annu. Wheat Newsl.* **66**, 116–117 (2020).

Li, Y. H. et al. Dissection of a rapidly evolving wheat resistance gene cluster by long-read genome sequencing accelerated the cloning of *Pm69*. *Plant Commun.* <https://doi.org/10.1016/j.xplc.2023.100646>, in press.

L55 Please provide references and the exact number of NLR?

Reply: We have provided references and the exact number of NLR as suggested in lines 57-59 in revised manuscript.

L120 It seems that no Methods are provided for this experiment. Please correct.

Reply: Thanks for the suggestions. Transcriptome sequencing of *Pm13a* donor parent TA3575 and recipient parent CS inoculated with *Bgt* isolate E09 was described as Dong et al. (2020). We have provided additional information in Methods “RNA-Seq of *Pm13a* donor parent TA3575” section in lines 474-484 in revised manuscript.

L129 Could you please provide how the annotation was performed.

Reply: The core domain of STPK was predicted based on the conserved domain database from the National Center for Biotechnology Information (NCBI) (<https://www.ncbi.nlm.nih.gov/Structure/cdd/wrpsb.cgi>). We have provided the prediction of core domain of STPK in Results “High resolution mapping and cloning of *Pm13a*” section in lines 133-135 in revised manuscript.

L130 When you said “structurally resembling” does it means that you have look at the 3D structure, if yes please provide methods.

Reply: Apologies for the imprecise expression. “structurally resembling” here means possessing same domain architecture. The *STPK* gene possessed the same domain architecture as the vertebrate necroptosis mediator MLKL, thus *STPK* was re-designated as *AelMLKL*. We have updated the statement in lines 133-136 in revised manuscript.

L130 please provide spelling for MLKL.

Reply: MLKL first appeared in the fourth paragraph of Introduction section. Therefore, we didn't spell out “MLKL” here.

L134 Please mention that you identified 14 susceptible families after phenotyping with isolate E09.

Reply: We have revised as you suggested in lines 140-141 in revised manuscript.

L141 what is the Brace region, it is called domain in your figure but not colored.

Reply: The brace is the region (a piece of peptide chain) that connects N-terminal MLKL_NTD domain and C-terminal kinase domain. We have provided the color of the brace region (Forest: Brace) in Fig. 2b in revised manuscript.

L158 did you used the full length cDNA or the CDS as indicated in your Fig3. Please correct.

Reply: We performed transgenic expression using coding sequence (CDS) of *AelMLKL* driven by maize ubiquitin (*Ubi*) promoter into susceptible wheat Fielder. We have made the revisions as you suggested in line 168 in revised manuscript.

L160 I'm a bit confused with the sentence, do you have 14 or 20 T₀?

Reply: Thanks for the suggestions. We got a total of 20 T₀ plants, however, of which, only 14 plants were *AelMLKL* (*Pm13*) positive T₀ plants.

We have changed the wording to improve the clarity in lines 169-170 in revised manuscript: As a result, 14 of the 20 transgenic T₀ plants were confirmed to have *AelMLKL* by PCR analysis and conducted immune responses to *Bgt* isolate E09 (IT = 0).

L160 according to your Sup Fig 4 there are different levels of expression of *Pm13a* in your transgenic. Could you please mention if this impacts the resistance level or anything else on the plants.

Reply: Although these 14 different *Pm13a* transgenic lines displayed variable expression levels of *Pm13a*, all 14 transgenic lines were immune to powdery mildew (R, IT = 0). So, the expression level difference in those plants didn't impact their resistance levels. We have put this statement in lines 170-175 in revised manuscript.

L163 the Pm13a gene. Instead of locus.

Reply: We have revised as you suggested in line 177 in revised manuscript.

L170 How can you say genetically divergent as we do not any data regarding these isolates.

Reply: Thanks for the suggestions. Among the 36 *Bgt* isolates, nine *Bgt* isolates (GY-KDZ-1, HB-XX-WXZ-1, SQ-YC-YJZ, ZK-CHQ-LHZ-2, SL-FS-5, SQ-YC-CGS, ZK-HY-QLZ-1,

ZK-HY-QLZ-2 and KF-YLH) were collected from different regions of Henan province, China. Five *Bgt* isolates (E09, E20, E21, E23 and E31) were collected from Beijing, China. Three *Bgt* isolates (E18, E26 and E32) were collected from Guizhou province, China. *Bgt* isolate E05 was collected from Yunnan province, China. The remaining 18 *Bgt* isolates were provided by Prof. Pengtao Ma, College of Life Sciences, Yantai University. The collection locations of the 18 *Bgt* isolates were not clear, but they were single-spore derived *Bgt* isolates with different virulent spectrums collected from different cities in China. We have changed “genetically divergent” to “different virulent spectrum” and provided this information in Supplementary Table 10 in revised manuscript.

L198 200 what do you suggest here, because *Pm13* homologs involved in resistance to biotrophs have also a kinase domain with conserved residues, this means that the kinase is important for the resistance? Could you please explain.

Reply: According to Hanks et al. (1988), protein kinases rely on the Lys of the Val-Ala-Ile-Lys (VAIK) motif for interaction with the α and β phosphates of ATP to position ATP during phosphotransfer; on the Asp of the His-Arg-Asp (HRD) motif within the catalytic loop (subdomain VIb) to act as a catalytic residue; and on the Asp within the Asp-Phe-Gly (DFG) motif in the activation loop (subdomain VII) to bind Mg^{2+} to coordinate the β and γ phosphates of ATP. The kinase domain of *Pm13* presents all three conserved catalytic residues that are crucial for phosphoryl transfer activity based on the presence of key residues for kinase activity, indicating that *Pm13* might also be a functional protein kinase for resistance. Furthermore, eight missense mutation sites in kinase domain of *Pm13* led to loss of resistance to powdery mildew, suggesting that the kinase is necessary for regulating *Pm13*-mediated powdery mildew resistance responses.

Reference:

Hanks, S. K., Quinn, A. M., and Hunter, T. The protein kinase family: conserved features and deduced phylogeny of the catalytic domains. *Science* **241**, 42-52 (1988).

L201 you cannot say functional domains as they were not functionally validated.

Reply: We agree with the reviewer that it is incorrect to say the domain was functional unless they were validated by the experiments. We have deleted “functional” in line 222 in revised manuscript.

L224 it could be that the expression of PR genes is triggered without *Pm13* as CS and TA3575 backgrounds are completely different. So please adjust rephrase.

Reply: To reduce the influence of different genetic backgrounds on PR gene expression, we re-examined the transcript levels of six PR genes (*PR1*, *PR2*, *PR3*, *PR4*, *PR5* and *PR9*) in *Pm13* transgenic lines AeIMLKL-0E11 and AeIMLKL-0E14, together with susceptible control Fielder at different time points after inoculation with *Bgt* isolate E09. As presented in Supplementary Fig. 14 in revised manuscript, both *Pm13* transgenic lines and susceptible control Fielder showed low expression levels of PR genes before infection. However, after infection, the PR genes expression levels increased in *Pm13* transgenic lines significantly higher than in susceptible control Fielder. The expression patterns indicated that *Pm13* transgenic lines induced the defense response more rapidly than susceptible control Fielder. We have revised in line 269-277 in revised manuscript and complemented the transcript levels of six PR genes (*PR1*, *PR2*, *PR3*, *PR4*, *PR5* and *PR9*) in

Pm13 transgenic lines AeMLKL-0E11 and AeMLKL-0E14, and susceptible control Fielder in Supplementary Fig. 14.

L227 there should be a detailed material and method section on the PCR with diagnostic markers so that everyone can exactly reproduce your experiment and evaluate their materials. Could you please explain how you know that the 180 lines lack *Pm13*.

Reply: We have provided more details about the PCR with *Pm13* diagnostic markers in Methods “Validation of *Pm13* functional molecular markers” section in lines 622-634 in revised manuscript.

Pm13 functional markers AeMLKL-1 and AeMLKL-8 were developed based on *Pm13* sequence. These two functional markers were uniquely amplified in *Pm13* stocks including CS-*Ae. longissima* 3S^l#2(3B) disomic substitution line TA3575, CS-*Ae. longissima* T3S^l#1S-3BS.3BL recombinant R1B, CS-*Ae. longissima* T3S^l#1S-3DS.3DL recombinant R2B, the newly developed CS-*Ae. longissima* T3S^l#2S-3BS.3BL recombinant W12-3 but absent in 180 wheat lines, indicating that the 180 lines lack *Pm13*.

L241 I’m not sure to understand the link between the identification of kinase and the possibility of pyramiding, this could have been done before the identification of kinase?

Reply: We have updated the statement to “Recently resistance genes encoding kinase fusion proteins have expanded the repertoire of non-NLR genes in the Triticeae for resistance breeding.” in lines 291-293 in revised manuscript.

L247 a most likely functional it was not experimentally validated.

Reply: Thanks for the suggestions. We agree with you that whether the domains are functional must be validated by related experiments. Thus, we has toned down the statement as suggest by several reviewers, we have deleted the word “functional” in revised manuscript.

L254 I like this discussion about the possible resistance mechanism of PM13 but I think that it would great to compare with the mechanism recently identified for NBS-LRR genes and the resistosome. The structure of the gene also questions me on how the gene will recognize the pathogen as it does not carry a domain like the LRR domain involved in recognition.

Reply: *Pm13* encodes a novel kinase fusion protein (KFP) that contains a kinase domain in its C terminus fused to a MLKL_NTD domain in N-terminal. According to the common model for KFPs (Yu et al. 2023, Wang et al. 2023, Fahima & Coaker 2023), the integrated domains in KFPs were hypothesized to perceive pathogen effectors, while the kinase catalyze the phosphorylation of either the effector or the NLR guard to initiate downstream defense responses and immunity. However, the *Pm13* MLK_NTD domain superimposed well with HeLo domain of animal MLKL and DUF1221 domain of *Arabidopsis thaliana* AtMLKL, all these domains-containing proteins conducted cell death, indicating that MLK_NTD domain of *Pm13* may function as inducing cell death. Thus whether the integrated MLKL_NTD domain present in *Pm13* may act as decoys of virulence effector proteins delivered into plant cells or regulate cell death remain unknown. We are also course about that how *Pm13* will recognize the pathogen, and decided to conduct experiments to explore the mechanism. We have discussed these in first and second paragraphs of Discussion section in revised manuscript.

Reference:

- Yu, G. T. et al. The wheat stem rust resistance gene *Sr43* encodes an unusual protein kinase. *Nat Genet.* **55**, 921-926 (2023).
- Wang, Y. J. et al. An unusual tandem kinase fusion protein confers leaf rust resistance in wheat. *Nat Genet.* **55**, 914-920 (2023).
- Fahima, T., & Coaker, G. Pathogen perception and deception in plant immunity by kinase fusion proteins. *Nat Genet.* **55**, 908-909 (2023).

L271 I would suggest removing broad spectrum as the mutant lines were evaluated only against one isolate.

Reply: We have deleted “broad spectrum” in revised manuscript.

L290 In this paragraph and the next one you discussed the difficulties of fine-mapping genes coming from wild species and accessions without reference genomes. While I understand the significance of the use of *ph* line in reducing introgression for its use in breeding programs, I am uncertain whether it is the optimal and fastest method for cloning genes from wild relatives, considering the current literature and the availability of new genomic tools. I would suggest elaborating on this.

Reply: Recently, several newly developed technologies facilitated the cloning of disease resistance genes in wheat and its wild relatives. Mutagenesis combined with NLR sequencing (MutRenSeq) or chromosome sequencing (MutChromSeq) allowed the rapid cloning of the resistance genes *Sr22*, *Sr45*, *Sr26*, *Sr61*, *Yr5*, *Yr7*, *Pm2*, *Rph1* and *Sr43*. Mutagenesis combined with mapping and RNA-Seq (MutRNA-Seq) or isoform sequencing (MutIsoSeq) facilitated the cloning of *Sr62*, *Lr9/Lr58*, *Pm69*, *YrNAM* and *Lr47*. In this manuscript, we cloned *Pm13* by combining *ph1b*-induced recombination and transcriptome sequencing of its direct donor parent TA3575. In comparison with other gene cloning approaches, our method for cloning a resistance gene from wild species introgressed into common wheat was probably not the optimal and fastest approach, however, it surpassed other methods for concurrently producing introgression lines with shortened alien segments harboring the gene for its direct use in breeding programs. We have complemented this information in the third and fourth paragraph of Discussion section in revised manuscript.

L301 This is not only the transcriptome but also the genetic mapping of contig issued from the transcriptome which is labor intensive. I suggest again here to compare with new strategies.

Reply: Thanks for the suggestions. We have complemented these comparison of gene cloning strategies in the third and fourth paragraph of Discussion section in revised manuscript.

L315 I was unable to find reference 67 but in reference 29 the durability of *Pm13* was mentioned based on a personal communication. While I acknowledge the importance of this communication, accurately assessing the long-term sustainability of this gene remains challenging based solely on this comment. I believe it is essential to exercise caution when discussing the durability of this gene and consider a more cautious approach. Especially as in the following sentence you mention that the gene was not deployed widely due to deleterious linkage drag. If it wasn't deployed largely it is not possible to really evaluate its durability.

Reply: Thanks for the suggestions. The reference 67 showed that *Pm13* was resistant to powdery mildew at the adult stage. The reference 67 (Guo et al. 2019) is changed to (Wu et al. 2018). Wu et al. 2018 displayed that *Pm13* was resistant to 39 *Bgt* isolates collected from different wheat fields located in the provinces of Hebei, Shandong, Henan, Beijing and Jiangsu, China. Nevertheless, we agree that resistance durability of *Pm13* only be confirmed by widely deployed in main commercial varieties. Hence, we have revised the statements as in lines 381-383 in revised manuscript: However, the gene had limited deployment in wheat breeding programs due to linkage drag resulting in inferior agronomic characteristics owing to the presence of a larger chromosomal segment of *Ae. longissima*.

Reference:

Wu, P. P. et al. Development of molecular markers linked to powdery mildew resistance gene *Pm4b* by combining SNP discovery from transcriptome sequencing data with bulked segregant analysis (BSR-Seq) in wheat. *Front. Plant Sci.* **9**, 95 (2018).

L327 functional !

Reply: We have revised the statement.

L337 Could you please provide the generations of the recombinants.

Reply: Thanks for the suggestions. CS-*Ae. longissima* recombinants T3S¹#1S-3BS.3BL (R1B) and T3S¹#1S-3DS.3DL (R2B) were developed by *ph1b*-induced homoeologous recombination (Ceoloni et al. 1988) and provided by Nanjing Agricultural University. R1B and R2B were homozygous recombinants. The generations are unsure, maybe at least 30 years.

Reference:

Ceoloni, C., Del Signore, G., Pasquini, M. & Testa, A. Transfer of mildew resistance from *Triticum longissimum* into wheat by *ph1*-induced homoeologous recombination. Proc 7th Int Wheat Genet Symp, Cambridge, UK, pp. 221–226 (1988).

L351 Could you please provide more information on powdery mildew isolate or provide a reference.

Reply: Among the 36 *Bgt* isolates, nine *Bgt* isolates (GY-KDZ-1, HB-XX-WXZ-1, SQ-YC-YJZ, ZK-CHQ-LHZ-2, SL-FS-5, SQ-YC-CGS, ZK-HY-QLZ-1, ZK-HY-QLZ-2 and KF-YLH) were collected from different regions of Henan province, China. Five *Bgt* isolates (E09, E20, E21, E23 and E31) were collected from Beijing, China. Three *Bgt* isolates (E18, E26 and E32) were collected from Guizhou province, China. *Bgt* isolate E05 was collected from Yunnan province, China. The remaining 18 *Bgt* isolates were provided by Prof. Pengtao Ma, College of Life Sciences, Yantai University. The collection locations of the 18 *Bgt* isolates were not clear, but they were single-spore derived *Bgt* isolates with different virulent spectrums collected from different cities in China. We have added more information in Supplementary Table 10.

L356 Please provide which generation of recombinants were phenotyped?

Reply: We used at least 16 individuals of F₂ progeny of each 3S¹#2 recombinant to evaluate resistance phenotype by inoculation with *Bgt* isolate E09. If all 16 F₂ plants derived from a recombinant were susceptible and half plants were positive for 3S¹-specific markers, the recombinant plant was susceptible phenotype. If 16 plants were segregating for resistance and

only the resistant plants were positive for 3S¹-specific markers, the recombinant plant was resistance phenotype. We have revised as you suggested in line 420 in revised manuscript.

P364 Provide the number of replicates.

Reply: All microscopic experiments were conducted with three biological replicates. Three leaves were collected as biological replicates and at least 100 infected cells per replicate were observed for detecting the ROS accumulation and cell death responses. We have provided this information in lines 438-441 in revised manuscript.

L376 please provide the names of the 4 markers

Reply: The four 3S¹#2-specific markers (CL54397, CL1543, CL90315 and CL88266) in the distal and proximal regions of each 3S¹#2 arm were used to analysis 1,580 selected individuals-derived BC₁F₂ progenies to select CS-*Ae. longissima* 3S¹#2 recombinants that missed 1-3 markers. We have revised in line 449 in revised manuscript.

L381 Please provide how markers were amplified and visualized.

Reply: As described previously, we have elaborated the concerned section and provided more details of PCR amplification in Materials “Initial mapping of *Pm13a* using CS *ph1b* mutant” section in lines 454-463 in revised manuscript.

L384 Could you please explain how the heterozygous recombinants were identified as your markers are dominant.

Reply: A total of 50 F₂ seeds from each 3S¹#2 resistant recombinant were sowed and the seedlings inoculated with *Bgt* isolate E09. If the 50 seedlings from a recombinant exhibited segregation for the resistant and susceptible, and the resistant and susceptible plants were 3S¹ marker positive and negative, respectively, the parental recombinant were heterozygous. The remaining seeds derived from heterzygous recombinants were used to constitute secondary segregating population for identification of novel 3S¹#2 recombinants for high-resolution mapping of *Pm13a*.

L411 Please provide the generation of the mutant used for sequencing and how cDNA were obtained.

Reply: M₃ homozygous mutants were used to amplify the full-length CDS sequence of *AelMLKL* gene for sequencing. The full-length CDS sequence were obtained as described in Method “Cloning of full-length CDS of *AelMLKL* gene” section.

L431 “with little homology in the wheat assemblies” What does it mean, please explain.

Reply: We have improved the clarity of this statement. It meant that the identity of the nucleotide sequences of target fragment was lower than 40% and the stretch of 100% nucleotide identity was less than 21 nucleotide sequences in comparison with common wheat Chinese Spring reference genome sequences. We have provided this information in lines 526-527 in revised manuscript.

L439 Could you please provide the light used for this experiment.

Reply: The detached leaves infected with *Bgt* isolate E09 were incubated at 24°C under a 14-h light / 10-h dark photoperiod. We have provided this information in line 535 in revised manuscript.

L441 15 plants were infected but if I understand only 3 were used for checking expression. Could you please confirm?

Reply: For BSMV-VIGS, 15 plants were infected and further used to check expression. Five plants for each biological replicate, and three biological replicates were used for each group.

L445 Could you please detail how the full length cDNA was cloned and the exact size/start/end of the full length cDNA.

Reply: The 1,789 bp length cDNA of *AelMLKL* was obtained from TA3575 transcriptome unigene CL897Contig1. The sequence of unigene CL897Contig1 was displayed in Supplementary Table 5. We performed transgenic expression using coding sequence (CDS) of *AelMLKL*. We have corrected it in line 541 in revised manuscript.

L462 Please mention how many leaves were included per biological replicates.

Reply: Each biological replicate consisted of five leaves from five individuals. We have supplemented this additional information in Methods “Gene expression analysis” section.

L473 Could you please provide further detail about the sequencing, the MasterMix used and the method of sequencing.

Reply: We have provided this information in Methods “Allelism at the *Pm13* locus” section in lines 577-584 in revised manuscript.

L488 Please provide the exact references for each genome. Could you please also define what you considered as *Pm13* homologs, what should be the minimum percentage of identity, and the length of alignment....

Reply: We searched for *Pm13* homologs across the whole-genomes of *T. aestivum* (cv. Chinese Spring), *T. urartu* (cv. G1812), *Ae. tauschii* (cv. AL8/78), *T. turgidum* dicoccoides (cv. Zavitan), *T. turgidum* durum (cv. Svevo), *Hordeum vulgare* (cv. Morex), *Secale cereale* (cv. Weining), *Thinopyrum elongatum* (cv. D-3458), and five Sitopsis species of *Aegilops* (*Aegilops bicornis* cv. TB01, *Aegilops longissima* cv. TL05, *Aegilops searsii* cv. TE01, *Aegilops sharonensis* cv. TH02, *Aegilops speltoides* cv. TS01) at WheatOmics (<http://202.194.139.32/blast/blast.html>) with default parameters. We discovered three *Pm13* orthologous genes in Sitopsis species (S genome) of *Aegilops* genus (TB01.3S01G0016800.1 from *Ae. bicornis* TB01, TH02.3S01G0013600.2 from *Ae. sharonensis* TH02, TS01.3B01G0036300.1 from *Ae. speltoides* TS01), which shared 99.37%, 97.27% and 89.94% sequence identity with *Pm13*, respectively. The *Pm13* orthologous genes reported in this manuscript had minimum 80% identity with *Pm13* and possessed the intact MLKL_NTD-Kinase domains.

Figure 1: Please provide the reference genome used to anchor the markers and calculate the size in Mb of the physical interval.

Reply: The markers sequences were blastn searched against *Aegilops longissima* TL05 reference sequences in Fig. 1.

Figure 2 c please indicate the time point

Reply: The third leaves from these representative plants subjected to VIGS were evaluated for powdery mildew 7 days after inoculation with *Bgt* isolate E09. We have provided these additional information in Fig. 2c legend in revised manuscript.

Figure 5 please provide the name of the statistical test. There is no description of the samples used in b, at which zedock stage roots were sampled? In which conditions? The same for all other samples. Please provide these details either here or in the Materials section. When using the delta delta Ct method, you need a control and treated samples. I suspect that you have in c, plants inoculated with *Bgt* strain and plant treated with water but what about in b? What are your controls?

Reply: Thank you for pointing this out. We complemented a series of additional experiments in Fig. 5. Fig. 5b and Fig. 5c have changed to Fig. 5d and Fig. 5e in revised manuscript. All data were analyzed by one-way ANOVA and multiple comparisons using Fisher's LSR test ($p < 0.01$) in Fig. 5d and Fig. 5e. We have supplemented this additional information in Fig. 5 legend in revised manuscript.

Transcript levels of *Pm13* were detected in different tissues of TA3575 including first leaves at three-leaf period (Z13), while roots, stems, flag leaves, glumes and spikes at heading stage (Z59). We have provided these additional information in Fig. 5d legend in revised manuscript.

Apologies for the imprecise expression. Expression analysis in Fig. 5d was calculated using $2^{-\Delta\Delta CT}$ method. We have corrected in Fig. 5 legend in revised manuscript.

Sup Fig2 please provide the reference genome used for physical position. Please indicate the isolate. And the generation of recombinants.

Reply: The physical position of 3S¹#2-specific molecular marker was blastn searched against *Ae. longissima* TL05 reference genome. *Bgt* isolate E09 was used for powdery mildew resistance evaluating the F₂ progenies of each of CS-*Ae. longissima* 3S¹#2 recombinants. We have provided this additional information in Supplementary Fig. 2 legend in revised manuscript.

Sup Fig3 Please the reference for the Ladder.

Reply: We have provided a reference for 100 bp Ladder DNA marker in Supplementary Fig. 3 as suggested in revised manuscript.

Sup Fig4 Could you please let us know if the T₁ plants used for *Pm13* expression were selected before by PCR for being positive. It is difficult to understand if the expression is based on 3 leaves from one experiment or one leaf from 3 different experiments. Please add in the legend how the relative expression was calculated.

Reply: The T₁ transgenic positive plants were selected by PCR using specific marker *AelMLKL_OE* before being used for *Pm13* expression analysis. The expression analysis is based on three biological replicates, five leaves from five positive individuals per replicate. We have provided this information in Supplementary Fig. 4 legend.

Sup Fig6 Could you please provide a reference for the 14 key residues of protein kinase. It would have be great to have AtMLKL in the alignment to see the non-conservation of some important residues.

Reply: We conducted a series of additional experiments, Supplementary Fig. 6 has changed to Supplementary Fig. 7 in revised manuscript. The 14 key residues of protein kinase were described by Hanks et al. (1988). We have AtMLKL in the alignment in Supplementary Fig 7 in revised manuscript. Eight of these 14 amino acids are found in kinase domain of AtMLKL1 and AtMLKL2, whereas seven out of the 14 were present in kinase domain of AtMLKL3.

Reference:

Hanks, S. K., Quinn, A. M. & Hunter, T. The protein kinase family: conserved features and deduced phylogeny of the catalytic domains. *Science* **241**, 42–52 (1988).

Sup Fig10 Please provide a reference for your marker or at least indicate the size of the bands on the gel.

Reply: We conducted a series of additional experiments, Supplementary Fig. 10 has changed to Supplementary Fig. 15 in revised manuscript. We have provided a reference for DL2000 Plus DNA marker and indicated the size of the bands on the gel in Supplementary Fig. 15 in revised manuscript. The size of specific amplified band for diagnostic functional marker AelMLKL-1 of *Pm13* is 399 bp. The size of specific amplified band for diagnostic functional marker AelMLKL-8 of *Pm13* is 936 bp.

Supplementary table 1: Please provide the version of the Chinese Spring genome and the reference for the *Ae. longissima* genome.

Reply: The version of the Chinese Spring genome was Chinese Spring RefSeq v2.1 and the reference for the *Ae. longissima* genome was *Ae. longissima* TL05 and AEG-6782-2 reference genome sequences. We have provided the information in revised manuscript.

Supplementary table 2: I guess there is an inversion between Markers and Materials name. The title indicates 44 markers whereas in the text it is 43. Please correct. Could you please explain what are R1B and R2B. and add which isolate was used.

Reply: Thanks for pointing out the discrepancies. We have corrected the number of 3S^l#2-specific markers in Supplementary Table 2. We developed 43 3S^l#2-specific markers based on transcriptome sequences of *Pm13a* donor parent TA3575.

R1B is CS-*Ae. longissima* T3BL.3BS-3S^l#1S recombinant carrying *Pm13* developed by Ceoloni et al. (1988). R2B is CS-*Ae. longissima* T3DL.3DS-3S^l#1S recombinant carrying *Pm13* developed by Ceoloni et al. (1988). All CS-*Ae. longissima* 3S^l#2 recombinants, R1B, R2B, CS and TA3575 were inoculated with *Bgt* isolate E09 to evaluate of powdery mildew resistance. We have provided this additional information in Supplementary Table 2.

Reference:

Ceoloni, C., Del Signore, G., Pasquini, M. & Testa, A. Transfer of mildew resistance from *Triticum longissimum* into wheat by *ph1*-induced homoeologous recombination. Proc 7th Int Wheat Genet Symp, Cambridge, UK, pp. 221–226 (1988).

Supplementary table 3: same comments as ST1.

Reply: We have provided the version of the Chinese Spring genome and the reference for the *Ae. longissima* genome. The 43 3S^l#2-specific molecular marker sequences were blastn searched against Chinese Spring RefSeq v2.1, *Ae. longissima* TL05 and AEG-6782-2 reference genome sequences.

Supplementary table 4: Could you please have it the same way as ST2. Same comments as ST2.

Reply: All CS-*Ae. longissima* 3S^l#2 recombinants, CS and TA3575 were inoculated with *Bgt* isolate E09 to evaluate powdery mildew resistance. We have provided this additional information in Supplementary Table 4.

Supplementary table 11: Please provide reference and version of the genomes. Are homologues present in syntenic position compared to *Pm13*?

Reply: We conducted a series of additional experiments, Supplementary Table 11 was changed to Supplementary Table 17 in revised manuscript. We have provided reference and version of the genomes in Supplementary Table 17 in revised manuscript. The synteny analysis was performed extending to five genes on either side of *Pm13* in *Ae. longissima* and compared this region across the Triticeae tribe presented in Supplementary Fig. 13 in revised manuscript. The synteny analysis showed that three homologues were present in syntenic position compared to *Pm13*.

Supplementary table 12: Could you please provide if these accessions are available form a genetic resource center.

Reply: This study was supported by the public funds and these accessions are freely available up request to the corresponding authors.

Reviewers' Comments:

Reviewer #2:

Remarks to the Author:

I thank the authors for performing a thorough revision of their manuscript. The additional phylogenetic and evolutionary analyses have significantly strengthened the manuscript. I also acknowledge that the authors tried their best to obtain the virulent powdery mildew isolates. It is a pity that the virulent isolates have not been maintained, as this would have allowed to unambiguously test the specificity of the Pm13 transgenic lines. Given the other analysis, including independent EMS mutants, gene silencing, and allele association studies, I do think that the authors provide sufficient evidence to support their conclusions of having cloned Pm13.

Reviewer #3:

Remarks to the Author:

The authors made several enhancements to the Pm13 manuscript, incorporating additional analyses and results, such as Pm13 protein localization, intraROS calculation, and evolutionary analysis. The newly included information contributes to a more comprehensive understanding of the role of Pm13 in conferring resistance to Bgt in wheat. However, we still believe that the manuscript requires further scientific and English editing before it can be accepted for publication. We have included some new comments and suggestions for further improvement of the manuscript:

1. Lines 32 to 36 - This sentence is too long. We propose the authors to break it into two sentences. The first should refer to the development of new introgression lines with reduced chromosome segments of *Ae. longissima* for deployment in bread wheat. The second should summarize the cloning of Pm13 and highlighting the important roles of KFPs in wheat immunity.
2. Line 58 - Please add MIW172 and MIWE18 to the list of the cloned NLRs.
3. Line 73 - We propose to change "We identified" to "Our previous study showed that".
4. Line 93-99 - This description of results is not clear. Thus, we suggest describing these results step by step, as follows: 1) The reaction of the two parental lines to Bgt; 2) The development of the segregating population and the ph1b-induced recombinants; 3) The genetic mapping using transcriptome data to identify SNPs in the Pm13 region.
5. Line 140-141 - We propose to add information on the generation of mutants, please indicate if these mutants are M1 or M2 generation.
6. Line 198 - We propose to change to "the resulting expression cassette (35::Pm13-GFP) was transiently expressed."
7. Line 200 - We propose to change "35::Pm13-GFP" to "Pm13-GFP" fusion protein. The format of gene and protein names should be checked along all the text. .
8. Line 211 - When writing "connection brace" do you mean a "linker"?
9. Line 227- We propose to change "the structural relationship" to "evolutionary relationships."
10. Line 253 - Please indicate if by 99.37%, 97.27% and 89.94% sequence identity you are referring to protein or DNA sequences.
11. Lines 272-277: The authors ignored our previous proposed explanation that "PR genes belong to the basal resistance responses; thus, a more logical explanation is that fungal effectors probably suppressed the expression of the PR genes in the susceptible line, rather than Pm13 is inducing the expression of PR genes"
12. Line 283, The 0 number does not make sense in "ranging from 0 to 2.82 Mb (3SI-31B)" sentence, since the approximate chromosome segments are estimated by reference genomes, please clarify.
13. Line 306, The mutant analysis can not prove that Pm13 has a protein kinase activity, it is only proving that the kinase domain is essential for Pm13 activation of the immune responses. Therefore, the conclusion that "Resistance loss induced by ten mutant sites from nine mutants located in the kinase domain of Pm13 implies that it functions as a protein kinase" should be changed.
14. Line 340 - We suggest to mention that "Long-read genome sequencing facilitated the cloning of Yr27 and Pm69".

15. Line 395 - We propose to change to "the resistant introgression line harboring Pm13 within approximately 2.82 Mb of 3Sl#2 genomic length (estimated using XX reference genome) and the developed diagnostic functional markers..."

16. Line 429-442 - There is no need to provide so many technical details when you cite a published method.

17. Line 588-589 - Gene names should be in italics.

Reviewed by Tzion Fahima and Yinghui Li

Reviewer #4:

Remarks to the Author:

The authors have adequately addressed all comments and have significantly enhanced the quality of the manuscript. I noticed two minor additional points. As mentioned before, I believe it would be great to include in the main text the fact that the gene is dominant. It seems that there is a word missing L458 "the PCR products of what".

Reviewer #5:

Reviewer 1 raised three major concerns about the manuscript.

Reviewer's major concern #1. The identification of an MLKL encoding gene as causal for disease resistance has not been described to date. Previous work by Mahdi et al. (2020) identified the role of MLKL proteins in plant immunity and cell death processes, but their work was not based on natural genetic variation and only identified a small number of proteins described as MLKLs (typically 2 to 3 per species). A preliminary bioinformatic analysis of AtMLKL1, AtMLKL2, and AtMLKL3 found that the N-terminal domain is classified as a DUF1221, whereas Pm13 is classified as carrying an MLKL N-terminal domain (CDD cd21037). While the authors have used AlphaFold2 to infer the potential structural relationship of Pm13 and AtMLKL1, there **lacks a complete analysis of the presence of this domain structure in other plant species**. The CDD database (<https://www.ebi.ac.uk/interpro/entry/cdd/CD21037/>) suggests that over 1k proteins carry the cd21037 domain in grasses, indicating that this overall family may be larger than previous described. The current work does not sufficiently place the impact of the discovery within the context of gene family evolution (specifically, MLKL_NTD-PK). This work has several implications:

A. It appears that the original finding by Mahdi et al. underestimated the number of MLKLs in plants. Therefore, the authors **should characterize** this extended set of proteins that may belong to the MLKL family.

B. The existing phylogenetic tree built for the protein kinase domain is **insufficient**. **No bootstrapping has been performed** and the domains themselves have not been placed within the overall context of protein kinase domains in angiosperms function of the plant protein kinase superfamily” by Melissa D. Lehti-Shiu and Shin-Han Shiu for a protocol on classifying the protein kinase domain in Pm13.

C. It was unclear which data sets that the authors have investigated to identify the presence of Pm13 in the Triticeae. I suggest a Supplemental Table that documents the presence or absence in diverse data sets analyzed. I noted that Pm13 appears to be naturally varying in *Aegilops bicornis*, *Aegilops longissima*, and *Aegilops sharonensis*. I suggest the authors investigate publicly available RNAseq data sets in NCBI.

The authors' response: A. To explore the potential structural relationship between MLKLs identified by Mahdi et al. (2020) and Pm13, we identified an additional 1,435 MLKL_NTD domain-containing (CDD cd21037) proteins and 100 DUF1221 domain-containing proteins from

Poaceae and Arabidopsis. Among these, 714 MLKL_NTD-Kinase proteins (Supplementary Table 11) and 97 DUF1221-Kinase proteins (Supplementary Table 12) were identified, respectively. We constructed a phylogenetic tree using these 811 proteins, together with three AtMLKL proteins (AtMLKL1, AtMLKL2 and AtMLKL3) reported by Mahdi et al. (2020) and Pm13. The dendrogram revealed that these 815 proteins were classified into two distinct clusters: namely MLKL_NTD-Kinase cluster and DUF1221-Kinase cluster. The three AtMLKLs belong to DUF1221-Kinase cluster where AtMLKL1 and AtMLKL2 in subfamily I and AtMLKL3 in subfamily II, consistent with the Mahdi's results. Pm13 was classified into the MLKL_NTD-Kinase cluster, however, it was in a subfamily distinct from any other MLKL proteins (Supplementary Fig. 8). The amino acid sequences of 714 MLKL_NTD-Kinase proteins and 97 DUF1221-Kinase proteins were provided in Supplementary Tables 11 and 12 in revised manuscript. The phylogenetic tree based on the 815 proteins was provided in Supplementary Fig. 8 in revised manuscript.

My opinion: How were 1,435 MLKL_NTD and 100 DUF1221 proteins selected for the databases? How were they reduced to 714 MLKL_NTD and 97 DUF1221-Kinase proteins for further analyses? $811+3=814$? Was anything wrong with the sentence 'The three AtMLKLs belong to DUF1221-Kinase cluster where AtMLKL1 and AtMLKL2 in subfamily I and AtMLKL3 in subfamily II, consistent with the Mahdi's results'? Do the proteins not selected for further analyses belong to a different cluster? The authors did not answer the question Reviewer 1 asked about the number of MLKLs in plant species.

The authors' response: B. To investigate the evolutionary origin of the Pm13 protein kinase domain, we applied a Hidden Markov Model-based classification approach for protein kinases developed by Lehti-Shiu & Shiu (2012) to analyze the kinase domain of Pm13. The result showed that Pm13 kinase domain belongs to the DLSV (DUF26, SD-1, LRR-VIII and VWA) protein kinase subfamily in the RLK (receptor-like kinase)/Pelle family. Further, we found that at least 44 genes had more than 60% amino acid sequence similarity with Pm13 kinase domain in Poaceae. By phylogenetic analysis of kinase domains of Pm13 and those 44 homologs together with recently reported KFP genes (Supplementary Fig. 9), we found that the kinase domain of Pm13 and 44 amino acid sequence similar genes belong to the LRR_8B subfamily (cysteine-rich receptor-like kinases), which is the most frequent kinase subfamily found in KFPs.

My opinion: The authors conducted further analyses and provided the information, but they did not directly answer where and how Pm13 originated during its evolution. Please note that a gene does not have amino acids. It was a wrong statement that at least 44 genes had more than 60% amino acid sequence similarity with the Pm13 kinase domain in Poaceae. There were many minor errors like this one.

The authors' response: C. We performed a BLASTP analysis with Pm13 across the whole-genomes of common wheat *T. aestivum* (cv. Chinese Spring), *T. urartu* (cv. G1812), *Ae. tauschii* (cv. AL8/78), *T. Turgidum dicoccoides* (cv. Zavitan), *T. turgidum durum* (cv. Svevo), *Hordeum vulgare* (cv. Morex), *Secale cereale* (cv. Weining), *Thinopyrum elongatum* (cv. D-3458), and five *Sitopsis* species of *Aegilops* (*Ae. bicornis* cv. TB01, *Ae. longissima* cv. TL05, *Ae. searsii* cv. TE01, *Ae. Sharonensis* cv. TH02, and *Ae. speltoides* cv. TS01) at WheatOmics (<http://202.194.139.32/blast/blast.html>). Based on phylogenetic analysis and synteny analysis using five *Ae. longissima* genes on either side of Pm13 across the Triticeae tribe, we found three Pm13 orthologous genes with intact MLKL_NTD-Kinase domain (TB01.3S01G0016800.1 from *Ae. bicornis* TB01, TH02.3S01G0013600.2 from *Ae. sharonensis* TH02 and

TS01.3B01G0036300.1 from *Ae. speltoides* TS01) that were present only in the S-genome of *Aegilops* species in the Triticeae tribe. Further, we downloaded the RNA-seq raw reads of the *Aegilops* species, including three *Ae. longissima* accessions (KU-14635, KU-14624, and KU-5752), two *Ae. bicornis* accessions (KU-14613 and KU-5784), three *Ae. sharonensis* accessions (KU-14668, KU-14663, and KU-14661), four *Ae. searsii* accessions (KU-14651, KU-6143, KU-6142, and KU-5755), seven *Ae. speltoides* accessions (KU-7848, KU-7716, KU-2236, KU-12963a, KU-14605, KU-14601, and KU-2208A) from the NCBI sequence read archive (SRA) database. Sequence comparison of the Pm13 sequence and RNA-seq raw reads of the *Aegilops* species, we found only *Ae. Longissimi* accession KU-14635 had identical cDNA sequence with Pm13. No reads or few reads from RNA-seq databases of other *Aegilops* species **demonstrated** an alignment on Pm13 cDNA sequence, suggesting likely lacking the **Pm13 allele** or weak expression of **Pm13 allele** in these reference accessions. Since we do not have these *Aegilops* species to test these possibilities, we have not added this information in the manuscript.

My opinion: It seems that the authors did not get the suggestion from Reviewer1 to directly tell whether Pm13 is absent or present in the publicly available databases. The authors repeatedly used ‘Pm13 alleles.’ It was not clear what the ‘allele’ means herein in plant species. An allele is one of two or more gene versions an individual inherits from each parent, such as a single base or a segment of bases. ‘Allele’ cannot be used for the orthologous genes among plant species. It was very challenging to judge if the author answered the question or not.

Reviewer’s major concern #2. The authors specifically state that the protein kinase domain of Pm13 is functional but they **provide no molecular evidence** for this claim (such as lines 200-202). The text either needs to **sufficiently temper** this statement (such as “**Pm13 is predicted to be a functional protein kinase based on the presence of key residues for kinase activity**”) or **kinase assays need to be performed**. Regardless, the authors need to prepare a supplementary figure where they show the position of the conserved residues for kinase activity.

The authors’ response: According to Hanks et al. (2008), protein kinases rely on the Lys of the Val-Ala-Ile-Lys (VAIK) motif to interact with the α and β phosphates of ATP to position ATP during phosphotransfer, the Asp of the His-Arg-Asp (HRD) motif within the catalytic loop (subdomain VIb) to act as a catalytic residue, and the Asp within the Asp-Phe-Gly (DFG) motif in the activation loop (subdomain VII) to bind Mg^{2+} to coordinate the β and γ phosphates of ATP. The kinase domain of Pm13 presents all three conserved catalytic residues that are crucial for phosphoryl transfer activity, thus we inferred that Pm13 kinase domain might also be a functional protein kinase based on the presence of key residues for kinase activity. The position of the conserved residues for kinase activity was presented **in Supplementary Fig. 7** in revised manuscript. However, we agree with reviewer’s suggestions that whether Pm13 kinase domain is functional or not should be proved by kinase function assays. Accordingly, we have updated the statement as suggested to “The C-terminal kinase domain of Pm13, in contrast to pseudokinase domains of the mammalian MLKL and *A. thaliana* AtMLKLs, presented all key conserved residues for kinase activity based on sequence alignment, **indicating** that Pm13 is likely a functional protein kinase.” in lines 218-220 in revised manuscript.

My opinion: What the authors revised somehow differed from what Reviewer 1 suggested in temper or sense. At least ‘indicating’ cannot be used in this case.

Reviewer's major concern #3. A statement about broad-spectrum resistance has to be tempered with the diversity of isolates used for assessment. In this work, the isolates are predominantly only from China (although I believe reference 29 tested a single Italian isolate), therefore I suggest the authors **temper their statements** and let the passage of time show the utility of Pm13.

The authors' response: According to Wu et al. (2018), Pm13 was resistant to 39 Bgt isolates collected from different wheat fields located in the provinces of Hebei, Shandong, Henan, Beijing and Jiangsu, China, indicating that Pm13 may be a putative broad-spectrum resistance gene. In the revised manuscript, we deleted "broad-spectrum".

My opinion: I have no comment on this one.

REVIEWER COMMENTS

Reviewer #2 (Remarks to the Author):

I thank the authors for performing a thorough revision of their manuscript. The additional phylogenetic and evolutionary analyses have significantly strengthened the manuscript. I also acknowledge that the authors tried their best to obtain the virulent powdery mildew isolates. It is a pity that the virulent isolates have not been maintained, as this would have allowed to unambiguously test the specificity of the *Pm13* transgenic lines. Given the other analysis, including independent EMS mutants, gene silencing, and allele association studies, I do think that the authors provide sufficient evidence to support their conclusions of having cloned *Pm13*.

We thank you for your thorough assessment of our manuscript. Your suggestions and comments have helped us improve the quality of the manuscript.

Reviewer #3 (Remarks to the Author):

The authors made several enhancements to the *Pm13* manuscript, incorporating additional analyses and results, such as Pm13 protein localization, intraROS calculation, and evolutionary analysis. The newly included information contributes to a more comprehensive understanding of the role of *Pm13* in conferring resistance to *Bgt* in wheat. However, we still believe that the manuscript requires further scientific and English editing before it can be accepted for publication. We have included some new comments and suggestions for further improvement of the manuscript:

Thank you for your feedback and positive assessment of our manuscript. The most recent version of the manuscript has been substantially revised incorporating all reviewers' suggestions. Further, a language expert has thoroughly edited the manuscript to enhance both its scientific content and English. We hope this manuscript now sends a clear message to the readers.

1. Lines 32 to 36 - This sentence is too long. We propose the authors to break it into two sentences. The first should refer to the development of new introgression lines with reduced chromosome segments of *Ae. longissima* for deployment in bread wheat. The second should summarize the cloning of *Pm13* and highlighting the important roles of KFPs in wheat immunity.

Response: We have incorporated the suggestion in lines 33-37 in the revised manuscript.

2. Line 58 - Please add *MIIW172* and *MIWE18* to the list of the cloned NLRs.

Response: We have added *MIIW172* and *MIWE18* to the list of the cloned NLRs as you suggested in line 61 in the revised manuscript.

3. Line 73 - We propose to change "We identified" to "Our previous study showed that".

Response: This has been changed as you suggested in lines 76-77 in the revised manuscript.

4. Line 93-99 - This description of results is not clear. Thus, we suggest describing these results step by step, as follows: 1) The reaction of the two parental lines to *Bgt*; 2) The development of the segregating population and the *ph1b*-induced recombinants; 3) The genetic mapping using transcriptome data to identify SNPs in the *Pm13* region.

Response: Thanks for this important suggestion. The section has been updated to improve clarity in lines 98-115 in the revised manuscript.

5. Line 140-141 - We propose to add information on the generation of mutants, please indicate if these mutants are M₁ or M₂ generation.

Response: To determine whether *AeMLKL* was required for *Pm13a* resistance, we mutagenized 4,800 seeds of TA3575 with 0.6% ethylmethanesulfonate (EMS), and 506 M₁ plants were harvested. Sixteen M₂ seedlings from each M₁ family were screened for susceptible mutants using the *Bgt* isolate E09, and twenty-five families segregating for resistance and susceptibility were identified and tested in the M₃ generation. Finally, 14 independent susceptible mutants were verified after powdery mildew resistance evaluation, 3S¹#2-specific marker analysis and *in situ* hybridization identification in the M₃ generation. We have added this information as you

suggested in lines 157-162 in the revised manuscript.

6. Line 198 - We propose to change to “the resulting expression cassette (35::Pm13-GFP) was transiently expressed.”

Response: The sentence has been updated as suggested in line 223 in the revised manuscript.

7. Line 200 - We propose to change “35::Pm13-GFP” to “Pm13-GFP” fusion protein. The format of gene and protein names should be checked along all the text.

Response: This change has been incorporated in line 225 in the revised manuscript.

8. Line 211 - When writing “connection brace” do you mean a “linker”?

Response: In this manuscript, “connection brace” means a “linker.”

9. Line 227- We propose to change “the structural relationship” to “evolutionary relationships.”

Response: We have changed as you suggested in line 253 in the revised manuscript.

10. Line 253 - Please indicate if by 99.37%, 97.27% and 89.94% sequence identity you are referring to protein or DNA sequences.

Response: In this manuscript, 99.37%, 97.27% and 89.94% sequence identity is referred to protein sequences. This has been mentioned in line 279 in the revised manuscript.

11. Lines 272-277: The authors ignored our previous proposed explanation that “PR genes belong to the basal resistance responses; thus, a more logical explanation is that fungal effectors probably suppressed the expression of the *PR* genes in the susceptible line, rather than *Pm13* is inducing the expression of *PR* genes”

Response: We apologize for missing the previous suggestion. The statement has been modified as you suggested in lines 303-304 in the revised manuscript.

12. Line 283, The 0 number does not make sense in “ranging from 0 to 2.82 Mb (3S1-31B)” sentence, since the approximate chromosome segments are estimated by reference genomes, please clarify.

Response: We have updated the statement as “We developed a T3S¹#2S-3BS.3BL recombinant W12-3, which has a small 3S¹#2 segment harboring *Pm13* in wheat background, the *Ae. longissima* 3S¹ chromosome segment length in the recombinant wheat chromosome was estimated to be approximately 2.82 Mb based on the *Ae. longissima* TL05 reference genome.” in lines 310-312 in the revised manuscript.

13. Line 306, The mutant analysis can not prove that Pm13 has a protein kinase activity, it is only proving that the kinase domain is essential for Pm13 activation of the immune responses. Therefore, the conclusion that “Resistance loss induced by ten mutant sites from nine mutants located in the kinase domain of Pm13 implies that it functions as a protein kinase” should be changed.

Response: We have changed the statement as “The resistance loss induced by nine mutant sites from eight mutants located in the kinase domain of Pm13 suggested that the kinase domain is

essential for Pm13-mediated activation of immune responses” in lines 334-336 in revised manuscript.

14. Line 340 - We suggest to mention that “Long-read genome sequencing facilitated the cloning of *Yr27* and *Pm69*”.

Response: We have supplemented this information in lines 371-372 in the revised manuscript.

15. Line 395 - We propose to change to “the resistant introgression line harboring Pm13 within approximately 2.82 Mb of 3S^l#2 genomic length (estimated using XX reference genome) and the developed diagnostic functional markers...”

Response: The statement has been modified in lines 429-431 in the revised manuscript.

16. Line 429-442 - There is no need to provide so many technical details when you cite a published method.

Response: We have updated the section as per suggestion in lines 466-473 in revised manuscript.

17. Line 588-589 - Gene names should be in italics.

Response: We have corrected the error in lines 619-620 as you suggested in revised manuscript.

Reviewed by Tzion Fahima and Yinghui Li

Reviewer #4 (Remarks to the Author):

The authors have adequately addressed all comments and have significantly enhanced the quality of the manuscript. I noticed two minor additional points. As mentioned before, I believe it would be great to include in the main text the fact that the gene is dominant. It seems that there is a word missing L458 “the PCR products of what”.

We thank you for your comments and meticulous assessment of the manuscript.

Response: As per your suggestion, we have included the information “the *Pm13* gene is dominant” in lines 75-76 in the revised manuscript.

Response: The PCR products without polymorphism were digested with restriction enzymes. We have updated the statement in lines 490-491 in the revised manuscript.

Reviewer #5 (Remarks to Authors):

Dr. An,

I have reviewed the rebuttal letter of Li et al. for their responses to questions and comments made by Reviewer 1 on the revised manuscript entitled "Wheat powdery mildew resistance gene *Pm13* from *Aegilops longissima* encodes a unique mixed lineage kinase domain-like protein" submitted to Nature Communications. As you suggested, I needed to review the rebuttal letter only. I investigated the detailed information in the text wherever I needed to understand the questions and answers better. I examined the detailed information in the text wherever I needed to understand the questions and answers better. My conclusion was that some questions were answered, but others were not. Many problems were caused by writing or English. If I were Reviewer 1, I would not be satisfied with the answers from the authors for the revision. Here is my report.

Response: We apologize for not adequately addressing some of the comments in the first revision. We appreciate your thorough review and have now addressed the comments and suggestions in our response below and updated the manuscript accordingly. Additionally, the manuscript has undergone comprehensive editing by a language expert to improve both its scientific content and English. We trust the revised manuscript conveys a clear message to the readers.

Reviewer 1 raised three major concerns about the manuscript.

Reviewer's major concern #1. The identification of an MLKL encoding gene as causal for disease resistance has not been described to date. Previous work by Mahdi et al. (2020) identified the role of MLKL proteins in plant immunity and cell death processes, but their work was not based on natural genetic variation and only identified a small number of proteins described as MLKLs (typically 2 to 3 per species). A preliminary bioinformatic analysis of AtMLKL1, AtMLKL2, and AtMLKL3 found that the N-terminal domain is classified as a DUF1221, whereas Pm13 is classified as carrying an MLKL N-terminal domain (CDD cd21037). While the authors have used AlphaFold2 to infer the potential structural relationship of Pm13 and AtMLKL1, there **lacks a complete analysis of the presence of this domain structure in other plant species**. The CDD database (<https://www.ebi.ac.uk/interpro/entry/cdd/CD21037/>) suggests that over 1k proteins carry the cd21037 domain in grasses, indicating that this overall family may be larger than previously described. The current work does not sufficiently place the impact of the discovery within the context of gene family evolution (specifically, MLKL_NTD-PK). This work has several implications:

A. It appears that the original finding by Mahdi et al. underestimated the number of MLKLs in plants. Therefore, the authors **should characterize** this extended set of proteins that may belong to the MLKL family.

B. The existing phylogenetic tree built for the protein kinase domain is **insufficient**. **No bootstrapping has been performed** and the domains themselves have not been placed within the overall context of protein kinase domains in angiosperms function of the plant protein kinase superfamily" by Melissa D. Lehti-Shiu and Shin-Han Shiu for a protocol on classifying the protein kinase domain in Pm13.

C. It was unclear which data sets that the authors have investigated to identify the presence of Pm13 in the Triticeae. I suggest a Supplemental Table that documents the presence or absence in

diverse data sets analyzed. I noted that Pm13 appears to be natural varying in *Aegilops bicornis*, *Aegilops longissima*, and *Aegilops sharonensis*. I suggest the authors investigate publically available RNAseq data sets in NCBI.

The authors' response: A. To explore the potential structural relationship between MLKLs identified by Mahdi et al. (2020) and Pm13, we identified an additional 1,435 MLKL_NTD domain-containing (CDD cd21037) proteins and 100 DUF1221 domain-containing proteins from Poaceae and Arabidopsis. Among these, 714 MLKL_NTD-Kinase proteins (Supplementary Table 11) and 97 DUF1221-Kinase proteins (Supplementary Table 12) were identified, respectively. We constructed a phylogenetic tree using these 811 proteins, together with three AtMLKL proteins (AtMLKL1, AtMLKL2 and AtMLKL3) reported by Mahdi et al. (2020) and Pm13. The dendrogram revealed that these 815 proteins were classified into two distinct clusters: namely MLKL_NTD-Kinase cluster and DUF1221-Kinase cluster. The three AtMLKLs belong to DUF1221-Kinase cluster where AtMLKL1 and AtMLKL2 in subfamily I and AtMLKL3 in subfamily II, consistent with the Mahdi's results. Pm13 was classified into the MLKL_NTD-Kinase cluster, however, it was in a subfamily distinct from any other MLKL proteins (Supplementary Fig. 8). The amino acid sequences of 714 MLKL_NTD-Kinase proteins and 97 DUF1221-Kinase proteins were provided in Supplementary Tables 11 and 12 in revised manuscript. The phylogenetic tree based on the 815 proteins was provided in Supplementary Fig. 8 in revised manuscript.

My opinion: How were 1,435 MLKL_NTD and 100 DUF1221 proteins selected for the databases? How were they reduced to 714 MLKL_NTD and 97 DUF1221-Kinase proteins for further analyses? $811+3=814$? Was anything wrong with the sentence 'The three AtMLKLs belong to DUF1221-Kinase cluster where AtMLKL1 and AtMLKL2 in subfamily I and AtMLKL3 in subfamily II, consistent with the Mahdi's results'? Do the proteins not selected for further analyses belong to a different cluster? The authors did not answer the question Reviewer 1 asked about the number of MLKLs in plant species.

Response: In this manuscript, we identified 1,435 proteins containing the MLKL_NTD domain by searching only the MLKL_NTD domain in *Poaceae* and *Arabidopsis* from the Interpro database. Additionally, 100 proteins containing the DUF1221 domain were identified by searching only the DUF1221 domain in *Poaceae* and *Arabidopsis* from the Interpro database.

Out of the 1,435 MLKL_NTD domain-containing proteins, 714 were found to be fusion proteins with the MLKL_NTD domain and kinase domain (**MLKL_NTD-Kinase**), while the remaining 721 proteins either solely contain the MLKL_NTD domain or have a fusion of the MLKL_NTD domain with a non-kinase domain.

Among the 100 DUF1221 domain-containing proteins, 97 were identified to contain a fusion of the DUF1221 domain with the kinase domain (**DUF1221-Kinase**), one protein exclusively contained the DUF1221 domain, and two proteins contained a fusion of the DUF1221 domain with a non-kinase domain.

We constructed a phylogenetic tree using 815 proteins. These 815 proteins included 714 MLKL_NTD-Kinase proteins, 97 DUF1221-Kinase proteins, three AtMLKL proteins (AtMLKL1, AtMLKL2 and AtMLKL3) reported by Mahdi et al. (2020) and the Pm13 protein identified in this manuscript ($714 + 97 + 3 + \text{Pm13} = 815$).

Phylogenetic analysis of these 815 proteins revealed that the three AtMLKL proteins were classified in the DUF1221-Kinase cluster in which AtMLKL1 and AtMLKL2 were in subfamily I and AtMLKL3 was in subfamily II, consistent with Mahdi's phylogenetic analysis result. While the Pm13 protein is classified into the MLKL_NTD-Kinase cluster but in a subfamily distinct from other MLKL_NTD-Kinase protein subfamilies in the MLKL_NTD-Kinase cluster.

There are totally 3,691 MLKL_NTD domain-containing proteins in plant species by searching only the MLKL_NTD domain from the Interpro database.

We have updated the text in the revised manuscript to avoid any confusion on the proteins used for phylogenetic analysis.

The authors' response: B. To investigate the evolutionary origin of the Pm13 protein kinase domain, we applied a Hidden Markov Model-based classification approach for protein kinases developed by Lehti-Shiu & Shiu (2012) to analyze the kinase domain of Pm13. The result showed that Pm13 kinase domain belongs to the DLSV (DUF26, SD-1, LRR-VIII and VWA) protein kinase subfamily in the RLK (receptor-like kinase)/Pelle family. Further, we found that at least 44 genes had more than 60% amino acid sequence similarity with Pm13 kinase domain in Poaceae. By phylogenetic analysis of kinase domains of Pm13 and those 44 homologs together with recently reported KFP genes (Supplementary Fig. 9), we found that the kinase domain of Pm13 and 44 amino acid sequence similar genes belong to the LRR_8B subfamily (cysteine-rich receptor-like kinases), which is the most frequent kinase subfamily found in KFPs.

My opinion: The authors conducted further analyses and provided the information, but they did not directly answer where and how Pm13 originated during its evolution. Please note that a gene does not have amino acids. It was a wrong statement that at least 44 genes had more than 60% amino acid sequence similarity with the Pm13 kinase domain in Poaceae. There were many minor errors like this one.

Response: We generated phylogenetic trees based on the kinase domain of Pm13 in Poaceae and observed that the kinase domain of Pm13 belonged to the LRR_8B subfamily, which is the most prevalent kinase subfamily found in KFPs. However, this phylogenetic analysis could not provide enough information to determinate the origin and evolutionary history of Pm13.

Additionally, we identified 220 MLKL_NTD domain-containing proteins across the Triticeae tribe in WheatOmics (<http://wheatomics.sdau.edu.cn/>) and constructed a phylogenetic tree based on these 220 proteins. The analysis revealed that Pm13 and its three orthologs from the S-genome of *Aegilops* species were clustered together. Therefore, we hypothesized that Pm13 might originate after the formation of the S-genome. However, we understand much more evidence is needed to answer the question of how *Pm13* originated during its evolution. Studying the evolution of *Pm13* could be an interesting follow-up study.

Thank you for highlighting the error of our statement. This information has been updated in lines 265-266 in the revised manuscript to state that "at least 44 proteins in *Poaceae* shared more than 60% sequence identity with the Pm13 kinase domain".

The authors' response: C. We performed a BLASTP analysis with Pm13 across the whole genomes of common wheat *T. aestivum* (cv. Chinese Spring), *T. urartu* (cv. G1812), *Ae. tauschii* (cv. AL8/78), *T. Turgidum* dicoccoides (cv. Zavitan), *T. turgidum* durum (cv. Svevo), *Hordeum vulgare* (cv. Morex), *Secale cereale* (cv. Weining), *Thinopyrum elongatum* (cv. D-3458), and five

Sitopsis species of *Aegilops* (*Ae. bicornis* cv. TB01, *Ae. longissima* cv. TL05, *Ae. searsii* cv. TE01, *Ae. Sharonensis* cv. TH02, and *Ae. speltoides* cv. TS01) at WheatOmics (<http://202.194.139.32/blast/blast.html>). Based on phylogenetic analysis and synteny analysis using five *Ae. longissima* genes on either side of *Pm13* across the Triticeae tribe, we found three *Pm13* orthologous genes with intact MLKL_NTD-Kinase domain (TB01.3S01G0016800.1 from *Ae. bicornis* TB01, TH02.3S01G0013600.2 from *Ae. sharonensis* TH02 and TS01.3B01G0036300.1 from *Ae. speltoides* TS01) that were present only in the S-genome of *Aegilops* species in the Triticeae tribe. Further, we downloaded the RNA-seq raw reads of the *Aegilops* species, including three *Ae. longissima* accessions (KU-14635, KU-14624, and KU-5752), two *Ae. bicornis* accessions (KU-14613 and KU-5784), three *Ae. sharonensis* accessions (KU-14668, KU-14663, and KU-14661), four *Ae. searsii* accessions (KU-14651, KU-6143, KU-6142, and KU-5755), seven *Ae. speltoides* accessions (KU-7848, KU-7716, KU-2236, KU-12963a, KU-14605, KU-14601, and KU-2208A) from the NCBI sequence read archive (SRA) database. Sequence comparison of the *Pm13* sequence and RNA-seq raw reads of the *Aegilops* species, we found only *Ae. longissima* accession KU-14635 had identical cDNA sequence with *Pm13*. No reads or few reads from RNA-seq databases of other *Aegilops* species demonstrated an alignment on *Pm13* cDNA sequence, suggesting likely lacking the *Pm13* allele or weak expression of *Pm13* allele in these reference accessions. Since we do not have these *Aegilops* species to test these possibilities, we have not added this information in the manuscript.

My opinion: It seems that the authors did not get the suggestion from Reviewer1 to directly tell whether *Pm13* is absent or present in the publicly available databases. The authors repeatedly used ‘*Pm13* alleles.’ It was not clear what the ‘allele’ means herein in plant species. An allele is one of two or more gene versions an individual inherits from each parent, such as a single base or a segment of bases. ‘Allele’ cannot be used for the orthologous genes among plant species. It was very challenging to judge if the author answered the question or not.

Response: Thanks for pointing it out. The term ‘*Pm13* alleles’ has been updated to ‘*Pm13* orthologous genes’. To date, neither the *Pm13* gene nor its protein sequence is present in the publicly available databases. However, some RNA-seq raw reads of *Ae. longissima* accession KU-14635 have shown alignment with the *Pm13* cDNA sequence in the NCBI sequence read archive (SRA) database. Besides, three *Pm13* orthologous genes (TB01.3S01G0016800.1 from *Ae. bicornis* TB01, TH02.3S01G0013600.2 from *Ae. sharonensis* TH02 and TS01.3B01G0036300.1 from *Ae. speltoides* TS01) could be found in the S-genome of *Aegilops* species within the Triticeae tribe by blasting *Pm13* DNA sequence at WheatOmics (<http://202.194.139.32/blast/blast.html>).

Reviewer’s major concern #2. The authors specifically state that the protein kinase domain of *Pm13* is functional but they provide no molecular evidence for this claim (such as lines 200-202). The text either needs to sufficiently temper this statement (such as “*Pm13* is predicted to be a functional protein kinase based on the presence of key residues for kinase activity”) or kinase assays need to be performed. Regardless, the authors need to prepare a supplementary figure where they show the position of the conserved residues for kinase activity.

The authors’ response: According to Hanks et al. (2008), protein kinases rely on the Lys of the Val-Ala-Ile-Lys (VAIK) motif to interact with the α and β phosphates of ATP to position ATP

during phosphotransfer, the Asp of the His-Arg-Asp (HRD) motif within the catalytic loop (subdomain VIb) to act as a catalytic residue, and the Asp within the Asp-Phe-Gly (DFG) motif in the activation loop (subdomain VII) to bind Mg²⁺ to coordinate the β and γ phosphates of ATP. The kinase domain of Pm13 presents all three conserved catalytic residues that are crucial for phosphoryl transfer activity, thus we inferred that Pm13 kinase domain might also be a functional protein kinase based on the presence of key residues for kinase activity. The position of the conserved residues for kinase activity was presented in Supplementary Fig. 7 in revised manuscript. However, we agree with reviewer's suggestions that whether Pm13 kinase domain is functional or not should be proved by kinase function assays. Accordingly, we have updated the statement as suggested to "The C-terminal kinase domain of Pm13, in contrast to pseudokinase domains of the mammalian MLKL and *A. thaliana* AtMLKLs, presented all key conserved residues for kinase activity based on sequence alignment, indicating that Pm13 is likely a functional protein kinase." in lines 218-220 in revised manuscript.

My opinion: What the authors revised somehow differed from what Reviewer 1 suggested in temper or sense. At least 'indicating' cannot be used in this case.

Response: We have rephrased this statement to 'Pm13 is predicted to be a functional protein kinase based on the presence of key residues for kinase activity' in lines 245-246 in the revised manuscript.

Reviewer's major concern #3. A statement about broad-spectrum resistance has to be tempered with the diversity of isolates used for assessment. In this work, the isolates are predominantly only from China (although I believe reference 29 tested a single Italian isolate), therefore I suggest the authors temper their statements and let the passage of time show the utility of Pm13.

The authors' response: According to Wu et al. (2018), *Pm13* was resistant to 39 *Bgt* isolates collected from different wheat fields located in the provinces of Hebei, Shandong, Henan, Beijing and Jiangsu, China, indicating that *Pm13* may be a putative broad-spectrum resistance gene. In the revised manuscript, we deleted "broad-spectrum".

My opinion: I have no comment on this one.

Response: Thank you for your comments and positive assessment of the manuscript.

Reviewers' Comments:

Reviewer #3:

Remarks to the Author:

Our comments and suggestions were properly addressed in the revised manuscript, and we recommend it for publication.

Tzion Fahima and Yinghui Li

Reviewer #5:

Remarks to the Author:

I found that the manuscript has been significantly improved in reading, as it has undergone comprehensive editing by a language expert in English, as the authors claimed. The authors have corrected the mistakes I found in the previous version. I do not think that the manuscript has been significantly improved in science because no additional experiments were done as suggested by Reviewer 1. However, understandably, the proposed experiments could not be done within a short period.